# Modeling Anthropogenic Aerosol Sources and Secondary Organic Aerosol Formation: A Wintertime Study in Central Europe

Hanna Wiedenhaus<sup>1</sup>, Roland Schrödner<sup>1</sup>, Ralf Wolke<sup>1</sup>, Marie L. Luttkus<sup>1</sup>, Shubhi Arora<sup>2</sup>, Laurent Poulain<sup>2</sup>, Radek Lhotka<sup>3</sup>, Petr Vodička<sup>3</sup>, Jaroslav Schwarz<sup>3</sup>, Petra Pokorna<sup>3</sup>, Jakub Ondráček<sup>3</sup>, Vladimir Ždímal<sup>3</sup>, Hartmut Herrmann<sup>2</sup>, and Ina Tegen<sup>1</sup>

**Correspondence:** Hanna Wiedenhaus (wiedenhaus@tropos.de)

#### Abstract.

Anthropogenic aerosol particles remain a significant air quality concern in Central Europe, particularly during winter months. This study employs the COSMO-MUSCAT chemistry transport model to investigate particulate matter sources, with a focus on emissions from residential heating. The model results are compared with winter measurements from one site in Germany and two sites in the Czech Republic, where solid fuels are commonly used for heating. A non-reactive tagging method tracking primary organic matter (OM) reveals a high contribution from residential heating. Although the magnitude and temporal changes of the model results mostly agree with total OM values at two measuring stations, it appears to underestimate measurements at a site in the central Czech Republic. This underestimation is partly attributed to the inadequate representation of secondary organic aerosol (SOA) emitted from wood combustion. The study highlights the impact of anthropogenic volatile organic compounds (AVOC) on SOA formation, which are currently underrepresented in air quality models. Sensitivity tests adjusting SOA yields and AVOC emissions increase OM concentrations of up to 40% at the measurement sites. These findings emphasize the need for accurate parameterization of AVOC derived SOA formation and residential heating emissions to better tackle wintertime air quality challenges in Central Europe.

## 1 Introduction

According to the European Environment Agency's air quality report, 238.000 premature deaths can be attributed to PM<sub>2.5</sub> (particulate matter of 2.5 μm or smaller aerodynamic diameter) exposure in the European Union (EU) in 2020 (EEA, 2022). In a review summarizing multiple decades of research, Anderson et al. (2011) emphasize that exposure to PM significantly increases the risk of cardiovascular and respiratory diseases, posing a major global public health challenge. A report by the World Bank Group (2022) estimates that the societal cost of ambient fine particulate matter pollution in the Europe and Central Asia region reached 4.6% of gross domestic product (GDP) in 2019. This estimate reflects the economic impact of PM<sub>2.5</sub> related health outcomes, including premature mortality, morbidity, and lost productivity. The target of the EU's Air Pollution Action Plan is a 55% reduction in premature mortality due to PM<sub>2.5</sub> by 2030 compared to 2005 levels (EEA, 2022). However, based

<sup>&</sup>lt;sup>1</sup>Modeling of Atmospheric Processes Department, Leibniz Institute for Tropospheric Research, Leipzig, Germany

<sup>&</sup>lt;sup>2</sup>Atmospheric Chemistry Department, Leibniz Institute for Tropospheric Research, Leipzig, Germany

<sup>&</sup>lt;sup>3</sup>Institute of Chemical Process Fundamentals, Czech Academy of Sciences, Prague, Czech Republic

on self-reported data, 19 EU Member States still fall at least 30% short of their 2030 PM<sub>2.5</sub> emission reduction targets in 2021. A study by Beloconi and Vounatsou (2023) found that as of 2021, 47.5% of Europeans were living in areas where annual mean PM<sub>2.5</sub> concentrations exceeded the new EU limit of 10 µg m<sup>-3</sup>, which will come into force in 2030. Ground level measurements of PM<sub>2.5</sub> from the European Air Quality Monitoring Network for 2021 and 2022 show a striking gradient between clean and polluted areas. Eastern European regions and the Po Valley in Italy have the highest annual mean concentrations, while central and western Europe have much lower PM levels (EEA, 2019). Simulations of PM<sub>2.5</sub> exposure and PM<sub>2.5</sub> related mortality for the year 2015 by Gu et al. (2023) also indicate higher concentrations and associated health risks in Eastern Europe.

In this transition zone between less and more polluted regions, the rural background station Melpitz in eastern Germany recorded the highest annual mean  $PM_{10}$  concentration in 2021 as reported by the 'European Monitoring and Evaluation Programme' (EMEP) (Fagerli et al., 2023). Previous studies in Germany have shown that long-range transport from Eastern Europe, particularly from combustion processes, is a major contributor to regional background particle concentrations (van Pinxteren et al., 2019, 2016). The inflow of air masses from the east was associated with  $PM_{10}$  concentration peaks leading to an increase in exceedances of the current daily limit value of 50  $\mu g$  m<sup>-3</sup> (van Pinxteren et al., 2019). However, the relative contributions of multiple combustion sources to primary and secondary paticles, as well as their transboundary transport remain insufficiently quantified. This needs to be better characterised to enable effective and better targeted mitigation strategies to address the prevailing air quality challenges.

30

Source apportionment (SA) studies aim to link ambient concentrations of pollutants to their emission sources. Within chemical transport models (CTM), two main methods can be used to do this: the emission reduction impact method and the mass transfer method. The emission reduction impact method, or brute force approach, assesses how pollutant concentrations respond to specific emission changes (Thunis et al., 2019). An extreme case, the 'zero-out' method, sets emissions from selected sources to zero and estimates their maximum possible impact on ambient concentrations. This approach helps to assess the potential impact of emission reductions on air quality (Clappier et al., 2017). Despite its conceptual simplicity, this method is computationally intensive and the results are highly sensitive to the reference scenario chosen.

The mass transfer method, implemented in CTMs as the tagged species approach, estimates contributions from different source sectors and regions by tracing the mass transport of pollutants from emission sources to local concentrations (Thunis et al., 2019). In this method, new tracers are introduced for the pollutants of interest and labeled according to their emission sources, allowing them to be monitored throughout the model run (Kranenburg et al., 2013). This approach facilitates the study of source contributions across both spatial and temporal scales, with source definitions directly linked to the emission inventories used as model inputs (Mircea et al., 2020). Tagged species modules for particulate source apportionment are implemented in several chemistry transport models: e.g. in LOTUS-EUROS (Kranenburg et al., 2013), in DEHM (Brandt et al., 2013), PSAT (Particulate Matter Source Apportionment Technology) in CAMx (Yarwood et al., 2007), TSSA (Tagged Species Source Apportionment) (Wang et al., 2009) and ISAM (Integrated Source Apportionment Method) (Kwok et al., 2013) in CMAQ (US EPA Office of Research and Development, 2024). Tagging approaches are not designed to assess the effectiveness of mitigation measures or the impact of emission reductions because they do not consider indirect chemical effects (Thunis et al., 2019). However, they do provide a direct, additive source attribution of pollutant mass concentrations.

The TRACE project: 'Transport and Transformation of Atmospheric Aerosol over Central Europe with an Emphasis on Anthropogenic Sources', aims to develop a comprehensive understanding of the contribution of transported anthropogenic aerosols relative to local emissions, integrating expertise in synergistic measurement methods and modelling tools. As part of this effort, this study aims to improve the understanding of the interaction between dispersion and transformation processes by investigating an area of large PM<sub>2.5</sub> concentration gradients in Central Europe. Therefore, we implement a non-reactive tagged species approach into the online Eulerian chemical transport model COSMO-MUSCAT (Wolke et al., 2012). The tagging approach is applied to identify primary PM sources with a focus on winter combustion emissions. Online and offline measurements from an extensive campaign in 2021 are used to validate the simulations and to improve the understanding of the local air quality.

# 2 Observations and modeling

#### 2.1 Sampling sites

The TRACE winter campaign took place from 05 February 2021 to 24 March 2021 at three measurement sites in central Europe: two of the stations, Melpitz (DE) and Košetice (CZ), were already well established as part of ACTRIS (Aerosol, Clouds and Trace Gases Research Infrastructure) and EMEP (European Monitoring and Evaluation Programme), while the third (Frýdlant, CZ) was specifically installed for this project. The sites were selected to capture an important area of transition between polluted and less polluted regions in Central Europe (see Fig. 1).

The research observatory Melpitz (51.54° N, 12.93° E, 86 m a.s.l.) is located 50 km east of Leipzig, Germany, observing atmospheric background conditions in Central Europe. It has been operated by TROPOS for more than 30 years (Spindler et al., 2001; Poulain et al., 2011). The station is surrounded by grassland and flat agricultural land without any notable wind obstacles. About 60% of the time throughout the year, the prevailing wind direction is south-west. These air masses are of maritime origin and reach Melpitz after having crossed Western Europe and, in the immediate vicinity, the city of Leipzig. Easterly winds occur 17% of the time throughout the year, bringing dry continental air masses affected by long-range transport from Poland, Belarus, Ukraine, Slovakia, and the Czech Republic (Spindler et al., 2001, 2012, 2013).

The National Atmospheric Observatory Košetice (49.35° N, 15.05° E, 534 m a.s.l.) is situated 60 km south-east of the Prague metropolitan area in Czech Republic. There are several small settlements in the vicinity of the station, however, the district is one of the least populated in the country (Zíková and Ždímal, 2013). Surrounded mainly by agricultural land and some woodland, a medium-sized timber factory equipped with a biomass furnace is located 7.5 km from the site (Schwarz et al., 2016). In winter, air masses reaching the site predominantly originate from South-West (44%) passing over Central Europe (Pokorná et al., 2022). Similarly, Lhotka et al. (2025) observed that the contribution of continental air masses was higher in winter compared to other seasons, highlighting a distinct seasonal difference.

The Frýdlant temporary measurement site (50.94° N, 15.07° E, 366 m a.s.l.) was set up 2 km north of the centre of Frýdlant, Czech Republic, close to the Polish border. The station is located on the north-western edge of the Jizera Mountains and is

o surrounded by forests and farmland. The Turów Coal Mine, a large Polish open pit mine, is about 10 km south-west of the site. Lignite from the Turów mine is used to fuel the nearby Turów power station.

During the campaign period, the COVID-19 pandemic continued to affect Europe, with containment measures still in place. In Germany, there was a lockdown from 13 December 2020 to 3 March 2021. Non-essential businesses, schools, and childcare facilities were closed, and employees were required to work from home wherever possible. Essential services such as supermarkets, pharmacies, and healthcare facilities remained open. From 3 March 2021, restrictions were adjusted locally based on infection rates and other factors (BMG, 2023). In the Czech Republic, strict restrictions were in place until 11 April 2021 (Slabá, 2022). In Poland, a partial lockdown was enforced from 28 December 2020 to 14 February 2021. Some restrictions were eased on 1 January 2021, allowing shops in shopping centers and cultural institutions to reopen. However, on 20 March 2021, stricter measures were reintroduced until 9 April (A3M Global Monitoring GmbH, 2023).

#### 100 2.2 Measurement data




A multi-device setup for data acquisition was in place at all three stations. The data presented in this study were measured with the instruments listed in Table 1. Instrumentation included, aerosol mass spectrometer for the non-refractory near PM<sub>1</sub> chemical composition (organic, nitrate, sulfate, ammonium and non sea-salt chloride) and a multi-wavelength aethalometer for the equivalent black carbon (eBC) connected to a dry PM<sub>10</sub> inlet. The mass concentration of PM<sub>2.5</sub> was measured by gravimetric filter sampling using a Digitel high-volume aerosol sampler with pre-heated quartz fiber filters. Samples were collected for 12 hours, covering daytime (5:00 to 17:00 UTC) and nighttime (17:00 to 5:00 UTC). The filters were further analysed with a Sunset Lab thermal-optical transmittance (TOT) instrument according to the EUSAAR2 temperature protocol (Cavalli et al., 2010). Online Sunset OC-EC data are also available for Frýdlant and Košetice. Carbon parallel plate diffusion denuders were used to remove volatile organic compounds to prevent positive sampling artefacts caused by adsorption of gas phase organics onto the filter (Turpin et al., 2000). An ACSM (Aerosol Chemical Speciation Monitor) was used for aerosol mass spectrometry at Košetice, while AMS (Aerosol Mass Spectrometer) instruments were used at Melpitz and Frýdlant. Hereafter, we use AMS/ACSM to refer collectively to measurements from all three instruments deployed at the sites. The AMS/ACSM instruments measure total organic matter (OM), which we can compare directly with our model output, while the two Sunset instruments detect the organic carbon (OC). For better comparability, OC was converted to OM using an OM/OC ratio from literature. Poulain et al. (2011) estimated an OM/OC ratio of 1.64 at the Melpitz station in winter 2009, with almost no diurnal variation. For a winter campaign in 2020 in Košetice Pokorná et al. (2022) found a ratio of 1.51 ( $\pm$  0.36). In this study we have applied a factor of 1.6 to the conversion of all Sunset data. Polycyclic aromatic hydrocarbons (PAHs) were detected on the filters by gas chromatography-mass spectrometry (GC-MS). For a detailed description of the measurement campaign see Arora et al. (in preparation).

#### 120 **2.3** Model description

All simulations were carried out with the multiscale model system COSMO-MUSCAT. It consists of two online coupled components, the regional numerical weather forecasting model COSMO (COnsortium for Small scale MOdelling) (version 5.05,

**Table 1.** Measuring devices used at the three sites to obtain the data for this study.




| species                                                                                      | device                                    |         | resolution        |         | station  |          |  |
|----------------------------------------------------------------------------------------------|-------------------------------------------|---------|-------------------|---------|----------|----------|--|
|                                                                                              |                                           | time    | size              | Melpitz | Frýdlant | Košetice |  |
| OM 50 2-                                                                                     | Aerodyne HR-ToF-AMS                       | 2.5 min | $PM_1$            | Х       |          |          |  |
| OM, SO <sub>4</sub> <sup>2-</sup> ,<br>NO <sub>3</sub> -, NH <sub>4</sub> +, Cl <sup>-</sup> | Aerodyne c-ToF-AMS                        | 5 min   | $PM_1$            |         | X        |          |  |
|                                                                                              | Aerodyne ToF-ACSM                         | 5 min   | $PM_1$            |         |          | X        |  |
|                                                                                              | Sunset Lab OC-EC offline Aerosol Analyzer | 12 h    | PM <sub>2.5</sub> | X       | X        | X        |  |
| OC, EC                                                                                       | Sunset Lab OC-EC online Aerosol Analyzer  | 2 h     | $PM_1$            |         | X        |          |  |
|                                                                                              | Sunset Lab OC-EC online Aerosol Analyzer  | 4 h     | $PM_{2.5}$        |         |          | X        |  |
| eBC                                                                                          | Magee Scientific AE33                     |         | $PM_{10}$         | X       | X        | X        |  |
| PM                                                                                           | Digitel High Volume Aerosol Sampler       | 12 h    | PM <sub>2.5</sub> | X       | X        | X        |  |
| PAH                                                                                          | GC/MS                                     |         | PM <sub>2.5</sub> | х       | х        | х        |  |
| Anhydromonosaaaharidas                                                                       | Agilent HP 6890 gas chromatograph         | 10.1    | PM <sub>2.5</sub> | X       | v        | v        |  |
| Anhydromonosaccharides                                                                       | and HP 5973 mass selective detector       | 12 h    |                   |         | X        | X        |  |

Schättler et al., 2018) in conjunction with the air-chemistry transport model MUSCAT (MUltiScale Chemistry Aerosol Transport) (Wolke et al., 2012), developed at TROPOS. The model system is designed for aerosol-chemistry process studies and air quality applications at the regional scale (Hinneburg et al., 2008; Heinold et al., 2011; Tõnisson et al., 2021; Wolke et al., 2012), and participated in model intercomparisons such as the Air Quality Model Evaluation International Initiative (AQMEII; Im et al., 2015; Galmarini et al., 2021). COSMO is a nonhydrostatic atmospheric numerical weather forecasting model based on the primitive thermo-hydrodynamic equations describing compressible flow in a moist atmosphere. The atmospheric equations are solved based on a terrain-following grid with rotated coordinates (Schättler et al., 2018). The meteorological model provides all the necessary meteorological fields (e.g. wind, relative humidity, temperature) to MUSCAT, which then simulates the transport and chemical transformations in the atmosphere for different gas and particle phase species. Transport processes include advection and turbulent diffusion, while physical loss processes are characterised by dry and wet deposition (Wolke et al., 2012). COSMO and MUSCAT operate largely independently on separate grids and are coupled at each horizontal advection time step (every 15–80 seconds), allowing highly time-resolved meteorological input for the chemistry-transport model.

Anthropogenic emissions of atmospheric compounds are treated as prescribed point and gridded area sources. Emissions within Germany are provided by the GRETA database of the German Federal Environment Agency (UBA) (Schneider et al., 2016) for the year 2019 (resolution:  $1 \text{ km} \times 1 \text{ km}$ ). For European emissions outside Germany the CAMS-REG-v5 emission inventory for the year 2018 (resolution:  $6 \text{ km} \times 6 \text{ km}$ ) is used, provided by the Copernicus Atmosphere Monitoring Service (CAMS) (Kuenen et al., 2022). Emissions are treated according to the Gridded Nomenclature For Reporting (GNFR) (NFR-I, 2023), i.e. they are grouped into different emission sectors representing different source types (e.g. Public Power, Traffic; see Table 2). The temporal variation of emissions (daily, weekly and seasonal cycle) is accounted for by time profiles, which differ according to the emitting sector. These temporal profiles are largely based on those provided with the TNO\_MACC-II

inventory (Kuenen et al., 2014), with adjustments for livestock and agriculture emissions according to Skjøth et al. (2011). For GNFR sector C ("Other Combustion"), the temporal profile weight applied to the emission factor ranges from 0.37 to 2.54 over the study period (see Fig. A1 in the Appendix). Emissions are provided as aggregated totals for some pollutant groups, which we then break down into individual components. Primary particulate matter (PM) is split into elemental carbon (EC), primary organic matter (OM), sulphate (SO<sub>4</sub><sup>2-</sup>), sodium and other minerals. A further distinction is made between fine (< 2.5 μm) and coarse (2.5 - 10 μm) aerosol particles. Non-methane volatile organic compounds (NMVOC) emissions are divided into 23 different hydrocarbon groups. The splitting profiles for PM and NMVOC are based on different literature sources and are also provided by CAMS (Kuenen et al., 2022). In this study, country-specific splitting profiles (based on the year 2018) are applied to the overall emission input (see Tables A2 and A1 for GNFR C splitting factors).

The emission of biogenic VOC (BVOC) is based on Steinbrecher et al. (2009) and improved for extended land use categories according to Luttkus et al. (2022). The primary natural aerosol components are emitted online in COSMO-MUSCAT. The estimation of desert dust mobilization depends on soil texture and soil size distribution according to Tegen et al. (2002) and preferential source regions (Heinold et al., 2011; Schepanski et al., 2017) using the current wind fields and hydrological conditions provided by COSMO. The emission of sea spray aerosol is based on Barthel et al. (2019).





Natural fire emissions (e.g. EC, OM and primary  $PM_{2.5}$ ) are provided as point sources for the year 2021 by the Global Fire Assimilation System (GFAS) (Kaiser et al., 2012). These emissions are resolved into 24-hour mean values with a specific injection height for each point source.

Dry deposition is modelled using the resistance approach described by Seinfeld and Pandis (2006). Aerosol particles and trace gases are also removed from the atmosphere through wet deposition, subdivided into in-cloud and below-cloud scavenging. Both processes are parameterized by size-dependent particle capture efficiencies and corresponding gas uptake coefficients (Simpson et al., 2012).

To describe the gas-phase chemistry, an extended version of the Regional Atmospheric Chemistry Mechanism RACM-MIM2-ext (Karl et al., 2006; Stockwell et al., 1997; Karl et al., 2009) is used. The mass-based aerosol population is described using a hybrid bulk-bin scheme. It comprises 25 prognostic aerosol particle tracers, including primary PM<sub>2.5</sub> and PM<sub>10</sub>, primary OM, EC, sulphate, nitrate and ammonium, secondary organic aerosol (SOA), as well as six bins for sea salt and primary marine organic particles (diameter range: 0.01–10 μm) and five desert dust bins (0.2–48 μm).

Secondary inorganic aerosol is formed through reactions between ammonia and sulfuric or nitric acid, which are generated from the gaseous precursor species sulfur dioxide (SO<sub>2</sub>) and nitrogen oxides (NO<sub>x</sub>) (Hinneburg et al., 2008). The partitioning between the particle and gas phases depends on the ambient atmospheric temperature and humidity. The implementation of this particle/gas partitioning follows the equilibrium approach described by Galperin and Sofiev (1998), utilizing the methods proposed by Mozurkewich (1993).

The formation of SOA is described by the module SORGAM (Schell et al., 2001), extended to include additional biogenic volatile organic compound (BVOC) precursors from isoprene, monoterpene and sesquiterpene oxidation and highly oxygenated molecule (HOM) formation from all considered BVOCs (Luttkus et al., 2022). The module uses the two-product approach described by Odum et al. (1996), which splits each SOA product class –comprising reaction products from aromatic precursors,

alkanes, alkenes,  $\alpha$ -pinene, and limonene— into two pseudo-products. For each, the formation of low volatility products and their gas/particle partitioning is simulated.

Low volatility condensable products are formed through oxidation of organic precursor gases by the OH radical, the nitrate radical  $NO_3$  and ozone. The amount produced is determined by a product species (i) dependent stoichiometric coefficient ( $\alpha_i$ ) in the specific reaction of the chemical mechanism (Schell et al., 2001). Then the SOA mass resulting from gas-particle partitioning is calculated using a partitioning coefficient  $K_{om,i}$  for each low volatile product species following Pankow (1994). The partitioning coefficient depends on temperature and is influenced by the molecular weight and saturation vapor pressure of species i. Each pseudo-product consists of a gas phase and particle phase product with different  $\alpha_i$  and  $K_{om,i}$ . All information regarding the precursor VOCs, SOA class names in both the gas and particle phases, along with the reactions and stoichiometric coefficients can be found in Schell et al. (2001) and in the supplement of Luttkus et al. (2022). The total SOA yield (Y) resulting from the two previous steps can be calculated according to the equation (1), where  $M_o$  is the total available absorbing organic matter (Odum et al., 1996).

$$Y = \sum_{i=1}^{n} Y_i = M_o \sum_{i=1}^{n} \left( \frac{\alpha_i K_{om,i}}{1 + K_{om,i} M_o} \right)$$
 (1)

Over a range of organic mass concentrations  $M_o$ , a precursor gas will have a range of aerosol yields Y. The relationship between yield and organic mass concentration can be determined through chamber measurements. To model this relationship, a curve is fitted by selecting the optimal values of  $\alpha_1$ ,  $\alpha_2$ ,  $K_{om,1}$  and  $K_{om,2}$  within the two-product framework. The sum of all particle phase products considered gives the total SOA concentration.

# 195 2.4 Model setup

180

185

205

The domains for the COSMO-MUSCAT simulations were chosen to cover the three measurement sites. To reduce computational costs for the targeted horizontal resolution in the measurement region, the model is nested twice. The innermost domain TraceD1 covers 317 × 204 grid cells with a horizontal resolution of ~2 km (see Fig. 1). The vertical resolution for COSMO in TraceD1 is 50 layers with a maximum height of 22 km, while MUSCAT uses only the lowermost 27 layers, i.e. up to ~6 km. A common grid nesting approach is used for the inner domains. The results of the larger domains are used as lateral boundary conditions on the inner domains. The meteorological initial and boundary conditions for the European domain (N0) are provided by reanalysis data of the CAMS global atmospheric composition forecasts (Inness et al., 2019). The simulation covers the period from 1 January to 31 March 2021, including a one month spin-up, with an output resolution of 1 hour. The model system is re-initialized every 48 h using the aerosol and trace gas concentrations at the end of the previous run and a 24 h COSMO pre-run to spin-up the meteorology in order to avoid long-term drifts in the modelled meteorology.

#### 2.5 Source attribution in COSMO-MUSCAT

A source attribution module has been developed for COSMO-MUSCAT 5.05 to analyse the influence of specific source regions, point sources, and emission sectors on primary particulate matter compounds. This new tagging method allows the individual

**Figure 1.** Domains for COSMO-MUSCAT runs and localization of the three rural background sampling sites (OpenStreetMap contributors, 2017).

tracking of emitted source-specific species during a single model simulation, thus enhancing the analytical capabilities of the model. Unlike the "zero-out" method, which requires multiple simulations for each source sector or region of interest, this new approach eliminates this need. As a result, the analysis is faster and less computationally intensive. To do so, an additional tracer is introduced into the model emissions for each species of interest from each defined source sector or source region, and combinations of both (see Table A3 in the Appendix). This additional tracer is labeled with the source information and then processed in parallel. In this way, the concentration of each of the so-called tagged tracers is available in each grid cell of the model and at each time step. This provides detailed spatial and temporal information about the source contribution to local tracer concentrations. In addition to the concentration of each tagged species, the total concentration - representing the cumulative impact of all sources - is also available. This allows the relative contribution of each tagged source to be effectively calculated. An overview of the selected source sectors is given in Table 2. Tags for source regions can be specified via a text-based input file in which each surface grid cell can be assigned a region name. For this study, we have tagged emissions from all countries within the inner domain TraceD1, namely Germany, Poland and the Czech Republic. Additionally, a source region 'Boundary' is introduced referring to the transport from the coarser domains to the inner domain. Input from outside the European domain is not included in the 'Boundary' tagged sector.

Transport (advection, diffusion, sedimentation) and removal (dry and wet deposition) processes are automatically applied to tagged tracers in the same way as for all other tracers. However, gas phase chemistry and aerosol chemistry are not considered at present within the tagging algorithm. Therefore, only chemically passive tracers can be tagged, i.e. non-reactive tagging approach. This enables a high spatial and temporal resolution analysis of the source composition of primary particles. As this study focuses on winter combustion processes, anthropogenic EC and OM emissions are tagged. EC and OM emissions are split into fine and coarse aerosol, therefore the same split is applied for the tagged tracers.

Table 2. GNFR source categories considered in this study.

| GNFR | Source Category      | Source Composition                                                                                                                                       |
|------|----------------------|----------------------------------------------------------------------------------------------------------------------------------------------------------|
| A    | Public Power         | Public electricity and heat production                                                                                                                   |
| В    | Industry             | Oil and gas refining, coal mining, iron and steel industry, chemical industry, pulp and paper industry, food and beverages industry, cement production   |
| C    | other Combustion     | Small combustion processes of private households, small businesses, agriculture, forestry and fishing                                                    |
| D    | Fugitives            | Fugitive emissions from oil and gas, exploration, production, transport and distribution of oil and natural gas                                          |
| F1   | Traffic: Gasoline    | Exhaust from gasoline powered vehicles                                                                                                                   |
| F2   | Traffic: Diesel      | Exhaust from diesel powered vehicles                                                                                                                     |
| F4   | Traffic: Non-Exhaust | Brake wear, tyre wear, gasoline evaporation and road wear                                                                                                |
| I    | Off Road             | Railways, off-road vehicles and other machinery, mobile combustion                                                                                       |
| K    | Livestock            | Enteric fermentation and manure management                                                                                                               |
| L    | Agriculture          | Application of manure and fertilizer, indirect emissions from managed soils, storage, handling and transport of agricultural products, use of pesticides |
|      | Other                | All other sectors are combined here: Product/solvent use, traffic: LPG/natural gas, shipping, aviation, waste treatment                                  |

#### 3 Results

#### 230 3.1 Meteorology

During the campaign notable meteorological events and sharp temperature changes occured in a short period of time. In early February, a low pressure system with cold air in the north and warm air in the south moved southwards, transporting cold air to the Balkans and Greece. On 7 and 8 February, strong easterly winds and heavy snowfall led to significant snow drifts in some areas of Central Europe. This was followed by a week of clear nights with prevailing westerly winds and temperatures dropping to -20°C. The model successfully captured the period of low temperatures and the snow event at all three stations (see Fig. 2). The snow event was followed by a cold episode resulting in more stagnant conditions with a change in wind direction and decreased wind speed at all stations.

In mid-February, a nearly stationary high pressure system transported warm air from the Sahara into Central Europe, driving a rapid temperature increase of up to  $20^{\circ}$ C within a week. An omega blocking pattern over Eastern Europe facilitated the inflow of dust that accompanied the warm air, allowing particles to travel as far north as Scandinavia (Hoshyaripour, 2021; Haarig et al., 2022). This event significantly affected all three stations, resulting in elevated surface dust concentrations of up to  $50 \mu g$  m<sup>-3</sup> (see Fig. 2). The unusually high, spring-like temperatures persisted until the end of February. Another significant dust event occurred on 3 March, originating from the Sahara and affecting Central Europe. Although our model successfully simulated dust uptake, surface concentrations during this event remained lower than those observed in mid-February. In mid-March, a shift to westerly winds brought low-pressure systems accompanied by widespread precipitation over Germany. This

Figure 2. Meteorological parameters for the three stations. The top row shows surface temperature and precipitation, with the shaded area representing surface dust concentration in the size class  $

Figure 3. Time series for  $PM_{2.5}$  mass concentration for the three stations. Filter data compared with modelled primary and secondary aerosol mass concentration. The timestamp for the filter data corresponds to the time of filter collection.

10.92 μg m<sup>-3</sup> for Frýdlant. Together with the NMB, the high RMSE indicate that the model tends to underestimate concentrations during periods of high concentration peaks, as the RMSE is particularly sensitive to outliers. All statistical parameters are presented in Table 3. Im et al. (2015) analysed the performance of multiple models in simulating PM<sub>2.5</sub> concentrations as part of the AQMEII model intercomparison project. They found that most models systematically underestimated PM<sub>2.5</sub> at rural stations, with biases ranging from -2% to -60%. The COSMO-MUSCAT model performed relatively well, showing a bias of -24.82%. However, all models struggled to capture wintertime levels, underestimating concentrations by more than 50% across all regions. During the first two weeks of February, the TRACE campaign revealed the largest discrepancies between observed

**Table 3.** Time-averaged measured and modelled mass concentrations and the associated Root Mean Squared Error (RMSE), Correlation Coefficient (R), Normalised Mean Bias (NMB) and fraction within a factor of 2 of the observations (FAC2) for the whole campaign period. The modelled data were adjusted to match the measurement intervals before statistical analysis. Online refers to in situ measurements and offline to filter sampling. A factor of 1.6 was applied to the OC measured by the Sunset offline instrument.

|                                 |          | model                            | online                           |                                           |      | offline |      |                                           |                                  |      |       |      |
|---------------------------------|----------|----------------------------------|----------------------------------|-------------------------------------------|------|---------|------|-------------------------------------------|----------------------------------|------|-------|------|
|                                 |          | mean                             | mean                             | RMSE                                      | R    | NMB     | FAC2 | mean                                      | RMSE                             | R    | NMB   | FAC2 |
|                                 |          | $[\mu \mathrm{g~m}^{\text{-3}}]$ | $[\mu \mathrm{g~m}^{\text{-3}}]$ | $[\mu \mathrm{g}~\mathrm{m}^{\text{-}3}]$ |      |         |      | $[\mu \mathrm{g}~\mathrm{m}^{\text{-3}}]$ | $[\mu \mathrm{g~m}^{\text{-3}}]$ |      |       |      |
| PM <sub>2.5</sub>               | Melpitz  | 6.80                             | -                                | -                                         | -    | -       | -    | 12.43                                     | 14.26                            | 0.25 | -0.45 | 0.48 |
|                                 | Košetice | 7.61                             | -                                | -                                         | -    | -       | -    | 17.24                                     | 13.85                            | 0.61 | -0.57 | 0.51 |
|                                 | Frýdlant | 8.17                             | -                                | -                                         | -    | -       | -    | 15.07                                     | 10.92                            | 0.34 | -0.46 | 0.6  |
| OM (AMS/ACSM PM <sub>1</sub> )/ | Melpitz  | 1.34                             | 1.59                             | 1.17                                      | 0.60 | -0.08   | 0.70 | 5.06                                      | 4.95                             | 0.24 | -0.73 | 0.17 |
| OM (offline PM <sub>2.5</sub> ) | Košetice | 1.66                             | 6.37                             | 6.49                                      | 0.39 | -0.74   | 0.21 | 7.74                                      | 8.12                             | 0.63 | -0.79 | 0.05 |
|                                 | Frýdlant | 1.81                             | 1.71                             | 2.01                                      | 0.19 | 0.18    | 0.49 | 6.18                                      | 5.18                             | 0.48 | -0.67 | 0.13 |
| eBC (AE33 PM <sub>10</sub> )/   | Melpitz  | 0.36                             | 1.00                             | 1.06                                      | 0.35 | -0.64   | 0.47 | 0.47                                      | 0.37                             | 0.29 | -0.23 | 0.64 |
| EC (offline PM <sub>2.5</sub> ) | Košetice | 1.00                             | 0.76                             | 0.66                                      | 0.50 | 0.30    | 0.56 | 0.41                                      | 0.70                             | 0.61 | 1.36  | 0.28 |
|                                 | Frýdlant | 1.06                             | 1.11                             | 0.93                                      | 0.45 | -0.07   | 0.66 | 0.44                                      | 0.88                             | 0.47 | 1.44  | 0.30 |
| sulfate                         | Melpitz  | 0.66                             | 0.62                             | 0.49                                      | 0.71 | 0.10    | 0.68 | -                                         | -                                | -    | -     | -    |
| (AMS/ACSM PM1)                  | Košetice | 0.73                             | 1.52                             | 1.38                                      | 0.36 | -0.51   | 0.46 | -                                         | -                                | -    | -     | -    |
|                                 | Frýdlant | 0.86                             | 0.78                             | 0.86                                      | 0.40 | 0.29    | 0.43 | -                                         | -                                | -    | -     | -    |
| nitrate                         | Melpitz  | 2.13                             | 1.58                             | 1.66                                      | 0.62 | 0.51    | 0.47 | -                                         | -                                | -    | -     | -    |
| (AMS/ACSM PM <sub>1</sub> )     | Košetice | 1.95                             | 2.65                             | 2.77                                      | 0.16 | -0.26   | 0.50 | -                                         | -                                | -    | -     | -    |
|                                 | Frýdlant | 2.03                             | 1.63                             | 2.15                                      | 0.46 | 0.40    | 0.37 | -                                         | -                                | -    | -     | -    |

and simulated PM<sub>2.5</sub> concentrations, with most other tracers also underestimated. Strong easterly winds until 8 February facilitated long-range pollutant transport to Melpitz and Frýdlant. The snow event on 7–8 February led to a decrease in PM<sub>2.5</sub> concentrations in Melpitz by approximately 10  $\mu$ g m<sup>-3</sup>. In Frýdlant, a slight decrease of around 4  $\mu$ g m<sup>-3</sup> was observed after the event, while in Košetice, concentrations even increased by about 4  $\mu$ g m<sup>-3</sup>, indicating limited overall washout effects. Concentrations rose again after the snow event, peaking around 10 February. The snow event was followed by a cold episode with stagnant conditions, reduced wind speeds, and a shift in wind direction, leading to pollutant accumulation. The model may underestimate residential emissions due to missing temperature dependencies and unaccounted COVID-19 lockdown effects. Increased heating activity due to unusually cold temperatures and limited mobility combined with stagnant meteorology could lead to the observed underestimation of PM<sub>2.5</sub>. Restricting the evaluation to data from February 15 onwards leads to improved model performance, with RMSE values of 9.64  $\mu$ g m<sup>-3</sup>, 12.44  $\mu$ g m<sup>-3</sup>, and 7.63  $\mu$ g m<sup>-3</sup> and corresponding NMB values of -10%, -52%, and -36% at Melpitz, Košetice, and Frýdlant, respectively. Additionally, the overall trend is better captured, with *R* increasing to 0.61 in Melpitz, 0.79 in Košetice and 0.65 in Frýdlant.

270

275

To gain a better understanding of the remaining discrepancies between modelled and measured  $PM_{2.5}$ , we can evaluate the accuracy for each individual  $PM_{2.5}$  component (see Figure 4).

**Figure 4.** Box and whisker plots for observations and modelled data at corresponding times during the campaign period. Rectangular boxes display the first and third quartiles. The line within each box represents the median value. Outliers are excluded. For OM and EC all data was averaged to 12 hours. For comparability with AMS/ACSM measurements, a factor of 1.6 was applied to the OC measured by the Sunset instruments.

# **Mineral Dust**

285

The Saharan dust outbreaks likely influenced the total PM<sub>2.5</sub> concentrations during the TRACE campaign. In the model, the February event brought high dust loads for several days and led to dust deposition at all three stations (see Fig. 2). Lidar measurements in Leipzig recorded pure dust conditions, but below 3 km height, aerosol from continental Europe was likely mixed into the Saharan dust plumes (Haarig et al., 2022). This event had a rather short travel time (less than two days) before reaching Leipzig. For the March event, the model also shows dust reaching the three stations, though the loads were not as high as during the second event. Observations by Haarig et al. (2022) detected mixed pollution-dust conditions after air masses

were transported over Spain and France, reaching Leipzig after 3-4 days. It is possible that the model underestimated surface  $PM_{2.5}$  during these events, potentially due to limitations in the model domain or insufficient vertical mixing to the surface.

# 290 Nitrate and Sulfate

295

300

At Melpitz, the model performs well for sulfate, with a correlation coefficient of 0.71 and a small bias (NMB =  $\pm$ 10%), while nitrate is overestimated (NMB =  $\pm$ 51%), though its temporal variability is reasonably captured (R = 0.62) (see Figure 4 panel (a) and (b)). At Frýdlant, the model shows moderate correlations (R  $\approx$  0.40 - 0.46) and biases (NMB =  $\pm$ 29% for sulfate and  $\pm$ 40% for nitrate) and a low agreement within a factor of 2 (FAC2 < 50%). Košetice exhibits the weakest agreement, with low correlations (R = 0.16 for nitrate, R = 0.36 for sulfate) and underestimations of both species (NMB =  $\pm$ 26% for nitrate and  $\pm$ 51% for sulfate). These results are broadly in line with model performance criteria reported in the literature, e.g., NMB within  $\pm$ 45% for sulfate and  $\pm$ 60% for nitrate (Huang et al., 2021), or NMB within  $\pm$ 30% and R > 0.40 (Emery et al., 2017). This indicates that the model reasonably captures the general magnitude and temporal variability of secondary inorganic aerosol concentrations across the domain, despite some site-specific discrepancies (Table 3). The AMS/ACSM may underestimate total sulfate and nitrate concentrations in winter, when particle growth shifts part of the mass beyond the PM<sub>1</sub> range (Poulain et al., 2020), though these species are generally predominantly found in PM<sub>1</sub> (Zhang et al., 2023). Given their relatively small contribution to total PM<sub>2.5</sub> at our sites, it is unlikely that secondary inorganic aerosols are responsible for the discrepancy between the predicted and measured PM<sub>2.5</sub> aerosol mass concentrations.

#### **Elemental Carbon**

EC concentrations show an overall good agreement with observations (see Fig. 4 (d)). Our model aligns more closely with 305 the Aethalometer data in Košetice (RMSE:  $0.66~\mu \mathrm{g}~\mathrm{m}^{-3}$ , NMB: +30%) and Frýdlant (RMSE:  $0.93~\mu \mathrm{g}~\mathrm{m}^{-3}$ , NMB: -7%) than in Melpitz, where it agrees well with the offline Sunset measurements (RMSE:  $0.37 \mu g m^{-3}$ , NMB: -23%). The discrepancy between Aethalometer and Sunset measurements arises from the different carbon fractions they detect: Aethalometers measure optically absorbing carbon (black carbon) in PM<sub>10</sub>, while Sunset instruments quantify elemental carbon (see Fig. A3 in the Appendix). Although differences in particle size cut-offs must be considered when comparing observations and model 310 results, Poulain et al. (2011) found that around 90% of the mass of elemental black carbon (eBC) in PM<sub>10</sub> is contained within the PM<sub>1</sub> fraction. Comparing across these different size classes should therefore be reasonable. In Košetice and Frýdlant, our model slightly overestimates EC concentrations, with NMB values of +136% and +144%, respectively. For winter 2019, Aethalometer measurements reported  $0.98 \pm 0.76 \ \mu g \ m^{-3}$  BC in Košetice (Lhotka et al., 2025), while Pokorná et al. (2022) found  $0.92 \pm 0.77 \,\mu \mathrm{g \, m^{-3}}$  for winter 2020. In comparison, our averaged model result for 2021 was 0.76  $\,\mu \mathrm{g \, m^{-3}}$ . In Melpitz, literature data show significant variability in BC concentrations. Atabakhsh et al. (2023) reported a value of 1.38  $\mu$ g m<sup>-3</sup> converted to PM<sub>1</sub> using a multi-angle absorption photometer (MAAP) during winter 2016/2017. Later, van Pinxteren et al. (2023) observed a marked decrease to  $0.5 \pm 0.41$  µg m<sup>-3</sup> in winter 2018/2019, likely reflecting reduced emissions and meteorological influences.

# 320 Organic Matter







The modelled OM that we refer to further is the sum of the fine primary organic aerosol (OM in  $PM_{2.5}$ ), the total SOA and OM from outside the European simulation domain. Primary OM accounts for approximately half of the total OM, with mean contributions of 44% in Melpitz, 48% in Frýdlant and a slighlty higher share of 57% in Košetice (see Fig. A2 in the Appendix). Panel (c) in Fig. 4 compares all available OM values for our campaign period. Across all three stations, the comparison to the Sunset data show a systematic underestimation by the model, with large negative NMB values: -73% in Melpitz, -79% in Košetice and -67% in Frýdlant. RMSE values are also high for Melpitz and Frýdlant (4.95 and 5.18  $\mu$ g m<sup>-3</sup>), but improve notably when considering only data from February 15 onwards, decreasing to 2.87 and 3.85  $\mu$ g m<sup>-3</sup>, respectively.

The underestimation of these values by our model seems to have a large contribution to the total PM<sub>2.5</sub> underestimation. The discrepancy between Sunset and AMS/ACSM observations may partly arise from the different particle size ranges each instrument targets: Sunset samples PM<sub>2.5</sub>, while AMS/ACSM captures only PM<sub>1</sub>. However, since organic aerosol is predominantly found in the submicrometer size range throughout the year (Poulain et al., 2020), the impact of the size cut-off on the comparison is expected to be minor. This is further supported by observations in Frýdlant, where both PM<sub>1</sub> (online) and PM<sub>2.5</sub> (offline) Sunset data are available and show only small differences. Nevertheless, other factors contributing to the observed discrepancy cannot be ruled out. AMS/ACSM instruments are particularly well suited for capturing temporal variability, due to their high time resolution. The Sunset instruments provide an estimate of the total carbonaceous mass and are useful for assessing the magnitude of concentrations. It uses the same filters as the gravimetric reference method, allowing a more direct comparison to total PM<sub>2.5</sub> mass and offering a more complete picture of the aerosol burden.

In Melpitz and Frýdlant, the model aligns reasonably well with AMS/ACSM observations, with RMSE values of 1.17 and 2.01  $\mu g \, m^{-3}$  and NMBs of -8% and +18%, respectively. Correlation is also relatively strong in Melpitz (R = 0.60), but lower in Frýdlant (R = 0.19), where the model fails to capture diurnal variability. The model underestimates the OM concentrations by AMS/ACSM in Košetice (RMSE: 6.48  $\mu g \, m^{-3}$ ; NMB: -74%) and also does not fully reproduce the diurnal variations (R = 0.39) (see Fig. A2 in the Appendix).

The correlation coefficient is good for Melpitz (0.60) indicating a good simulation also of the overall trend. While for the Sunset data, the correlation coefficient is only 0.24. The range of concentration of the Sunset data is underestimated by the model (NMB = -73%), which indicates that the model might be missing OM at this site. For Frýdlant, the diurnal patterns are not well met by the model (R in comparison with AMS = 0.19), while also the overall range of OM concentrations is underestimated (Sunset offline NMB = -67%). In Košetice the AMS/ACSM and both Sunset instruments give consistent results, while the modelled data is noticeably lower. The AMS/ACSM detects lower values than our model and also modelled diurnal patterns do not match the observation (NMB = -74%, R=0.39). In comparison to the Sunset filter measurements the model shows a similar underestimation (NMB = -79%). For Košetice, the same concentration levels for PM<sub>1</sub> and PM<sub>2.5</sub> size class OM indicate a dominance of fine aerosol, while there is few coarse mode organic aerosol. The correlation coefficient of the model concentrations against the AMS/ACSM measurement is lower (0.39) than that of the filter samples (0.63). If only the SOA components of the modelled OM are taken into account, the correlation coefficient compared to the AMS/ACSM

for Košetice decreases further to 0.18. For Frýdlant and Melpitz, calculating the correlation coefficient using only SOA gives similar results to using the total modelled OM concentration. The reduced correlation at Košetice when isolating SOA implies that the model underestimates secondary aerosol at this site, thereby negatively affecting the overall correlation.

Previous Sunset filter measurements taken at Melpitz in winter 2018/2019 found an averaged value of  $3.2 \pm 3.2~\mu g\,m^{-3}$  (van Pinxteren et al., 2023). AMS/ACSM data for winter 2009 also gives comparable values  $2.08 \pm 1.6~\mu g\,m^{-3}$  (Poulain et al., 2011) while measurements with an ACSM in winter 2016/2017 show higher values of  $6.21~\mu g\,m^{-3}$  (Atabakhsh et al., 2023). For Košetice, a good characterisation of the site is also given by various previous studies. AMS/ACSM measurements provide average values of  $3.13~\mu g\,m^{-3}$  in winter 2019 (Lhotka et al., 2025) and  $4.55 \pm 4.40~\mu g\,m^{-3}$  in winter 2020 (Pokorná et al., 2022). Mbengue et al. (2018) found an average OC concentration in  $PM_{2.5}$  of  $2.85 \pm 1.91~\mu g\,m^{-3}$  for the period 2013 - 2016. For our study period we found Sunset Filter values ranging in average from  $5.06~\mu g\,m^{-3}$  in Melpitz to  $7.74~\mu g\,m^{-3}$  in Košetice, exceeding typical values reported for previous years. This suggests a strong influence of meteorological conditions on the overall concentration levels.

The discrepancy between modelled and measured  $PM_{2.5}$  concentrations does not appear to be primarily driven by deviations in elemental carbon, sulfate, or nitrate concentrations. The overall good agreement between modelled and observed EC values, with correlation coefficients up to 0.61 and low bias, indicates a reliable simulation of primary combustion aerosol emissions. The contribution of secondary inorganic aerosol to total  $PM_{2.5}$  are limited and the discussed modeling uncertainties are likely not the main reason for the underestimation of total  $PM_{2.5}$ . OM is significantly underestimated, especially at Košetice (NMB = -74%, R = 0.39), which explains a large part of the  $PM_{2.5}$  model bias. We hypothesise that the underestimation of secondary organic aerosol is a major source of error in total  $PM_{2.5}$  simulations. A spatial variation in the model's performance is apparent, with similar trends observed in Melpitz and Frýdlant, whereas Košetice exhibits distinct behaviour. The dominance of fine particles in OM, suggested by nearly identical concentrations in the  $PM_1$  and  $PM_{2.5}$  size fractions, points to elevated levels of secondary particles. Therefore, the underestimation could indicate a general underrepresentation of SOA during winter in this area in COSMO-MUSCAT.

#### 3.3 Source attribution for elemental carbon and primary organic matter







Since the model accurately reproduces EC concentrations, which represent a primary anthropogenic aerosol component, we conclude that anthropogenic sources are well represented in the model, enabling reliable identification of source contributions. Additionally, with approximately half of the total OM comprised of POA, we infer that overall source profiles can be effectively identified by analysing primary OM and EC using the non-reactive tagging approach. The results, shown as relative contributions to primary OM and EC for the cold and warm period (Fig. 5), underline the importance of long-range transport of particles. The source sector 'Boundary' represents transported particles from the outer model domains into the innermost domain where tagging is applied. During the warm period, long-range transport accounts for about 38% of both EC and OM in Melpitz, illustrating the significant influence of particles originating outside the domain. In Košetice the contribution is 23.8% for OM and 22.8% for EC, while Frýdlant has the lowest influence with 16.6% for OM and 14.6% for EC, respectively. The prevailing wind regime and the basin-like topography of the Czech Republic reduce the influence of long-range transport at

Košetice compared to the other two stations. The 'Boundary' contribution to fine OM and EC is only slightly higher than for Frýdlant, which is located in the middle of the domain (see Fig. 8). Backward trajectory analyses (HYSPLIT; Stein et al., 2015) indicate that during the high PM peak event in early February, stationary meteorological conditions resulted in minimal air mass transport to all sites. This effect is particularly pronounced in Košetice, where strong local stagnation can be observed.




**Figure 5.** Relative source contributions to primary organic matter in PM<sub>2.5</sub> and elemental carbon in PM<sub>2.5</sub>. Top: cold period (05.02.2021 - 16.02.2021), bottom: warm period (16.02.2021 - 23.03.2021)

The study region, spanning parts of Germany, Poland, and the Czech Republic, is characterised by a high density of active lignite mines (see Fig. 6). Lignite, a particularly emissions-intensive fuel, is the energy source for many large power plants in this area. Germany and Poland host the largest number of coal-fired power plants with the highest total capacities in the EU (Alves Dias et al., 2018). Emissions from power plants used for electricity and heat production are categorized under the source sector 'Public Power'. Despite its proximity to areas with a high number of coal-fired power plants, the 'Public Power' sector contributes only a small share to the overall concentration of primary OM and EC. Tagging results for this sector, split by country of origin, are shown in Figure 7, indicating that the peaks at Frýdlant are predominantly driven by Polish emissions. The proximity of the Turów lignite power plant largely explains the observed peaks, especially during periods of low wind speeds. During other periods, emissions from German and Czech sources dominate. The influence of coal burning on air quality in Frýdlant is further amplified by its use in domestic heating. In 2017, 47.7% of Polish households with individual heating relied on coal (Macuk, 2019).

**Figure 6.** Spatial distribution of the average absolute contribution of emissions from the source sector Public Power to the concentrations of primary organic matter in  $PM_{2.5}$ . Areas with many coal-fired power plants are highlighted.

The sector 'other Combustion' includes combustion processes of private households, in particular domestic heating processes with all fuel types. This sector has the biggest contribution with up to 76.3% for EC and 72.6% for primary OM in Frýdlant. Contributions to fine OM from the 'other Combustion' sector are highest in the Czech Republic and in urban agglomerations in Poland and around Berlin, Germany (see Fig. 8, right panel). The main contributors to the concentrations observed at the stations are emissions originating within the country where the station is located. However, Melpitz stands out with the highest proportion of contributions from cross-border emissions. Atabakhsh et al. (2023) carried out a positive matrix factorization (PMF) analysis over a period of one year at Melpitz. They found the highest coal combustion contribution to POA under the influence of easterly continental air masses. Furthermore, they found a temperature and RH dependence for the factor consisting of aged SOA and highly oxidised OA in winter, with the highest concentrations observed at temperatures below 0°C and RH above 80%. They concluded that increased precursor emissions due to higher heating activities and amplified aqueous phase chemistry lead to increased SOA formation. This could suggest a potential additional underestimation of the SOA formation rate in early February, as strong easterly winds were observed, followed by a subsequent cold period.

# 415 4 Discussion




Chen et al. (2022) conducted a multi-year PMF source apportionment study across various locations in Europe. They identified a coal combustion factor of primary OA at only two sites: Melpitz (data collected in 2016/2017) and the urban location Kraków (data collected in 2018). The strong seasonal variations in this factor suggest it originates from residential heating emissions. The study also examined Košetice, where no coal combustion factor was detected; however, biomass burning accounted for 15.5% of the total OA in winter 2019. Lhotka et al. (2025) conducted a PMF study with data also collected in 2019 in Košetice.

**Figure 7.** Elemental carbon in PM<sub>2.5</sub> concentration, broken down by country of origin and source sector. Left: 'Public Power', right: 'other Combustion'. The wind barbs represent wind direction and speed, with 12-hour averages modelled for each station at surface level.

They identified a coal combustion factor with the highest contribution of 5% to total OA in spring, while biomass combustion contributed most in winter (12% of total OA). Both factors showed similar diurnal cycles related to domestic heating, and a strong correlation between levoglucosan and the biomass combustion factor was observed in winter, indicating a high proportion of wood combustion. During a particularly cold period in January 2019, an increased contribution of coal was observed, probably due to its increased use in private households for heating, given its higher calorific value compared to wood. The results are also consistent with those of Horník et al. (2024), who performed a PMF study with samples collected during the TRACE campaign for water-soluble organic compounds using NMR. They found a high residential heating contribution with coal markers indicating additional coal combustion in early February in Košetice.


Pokorná et al. (2018) analysed changes in PM2.5 composition and sources from the 1990s to 2009/2010 in Košetice. During this period, the dominant sources shifted from lignite combustion by power plants and oil combustion to residential heating, mainly with coal and/or biomass. In the Czech Republic only 5% of total coal consumption in 2019 was used in the residential sector, as part of the 'other combustion' source sector (IEA, 2021). Hovorka et al. (2015) conducted a receptor modelling study in a residential area 64 km north-east of Prague in winter 2013, and estimated that wood burning contributed 49% to the mass

**Figure 8.** Spatial distribution of the average absolute contribution of emissions from source sectors 'other Combustion' (top) and 'Boundary' (bottom) to the concentrations of primary organic matter in PM<sub>2.5</sub>.

of fine aerosol. They found high correlations between contributions from wood combustion and levoglucosan and suggested that wood combustion in local boilers is common in suburban areas in the Czech Republic.




The landscape surrounding Košetice is mainly agricultural with scattered woodland, the only direct sources of pollution are local roads and domestic heating (Zikova and Zdimal, 2016). It is plausible to assume high rates of wood burning, given the proximity of the timber factory. Levoglucosan, an aerosol tracer which is associated with biomass burning, measured during the TRACE campaign show highest mean concentrations in Košetice (0.32 µg m<sup>-3</sup>) and lowest in Melpitz (0.15 µg m<sup>-3</sup>). Polycyclic aromatic hydrocarbons (PAH) are also good tracers of combustion processes, e.g. retene is a unique marker of wood combustion (Ramdahl, 1983). The average retene concentration in Košetice is 2.13 ng m<sup>-3</sup> at average total PAH concentration of 24.43 ng m<sup>-3</sup>. In Frýdlant the averaged total PAH concentration is comparable (24.73 ng m<sup>-3</sup>), but retene concentrations are lower (1.01 ng m<sup>-3</sup>). Melpitz shows similar retene concentrations as Frýdlant (1.16 ng m<sup>-3</sup>) at lower total PAH levels (14.12 ng m<sup>-3</sup>). The high relative and absolute levels of retene and levoglucosan in Košetice are a good indicator for a high contribution of wood burning (Arora et al., in preparation). The results are also consistent with those of Horník et al. (2024), who reported high levels of levoglucosan in Košetice and Frýdlant. Overall, the results indicate a strong influence of wood burning for domestic heating during winter in the Košetice area. During particularly cold periods, residents appear to supplement wood with coal, leading to a greater local impact of coal emissions on air quality. The higher coal contributions observed

in Melpitz seem to be mainly driven by long-distance transport, whereas in Frýdlant, additional contributions from the nearby power plant are evident.

#### 4.1 Effects of COVID-19 containment measures






With containment measures still in place during the campaign, the daily lives of many of the region's citizens were disrupted. Different patterns of mobilization, the closure of businesses and changes in leisure habits are all factors that can affect air quality. The emission inventories used in this study do not take into account exceptional events affecting emissions, such as the COVID-19 restrictions. Several studies have looked at the impact of these restrictions on air quality. Most of them focus on the year 2020, when the pandemic peaked.

Gkatzelis et al. (2021) reviewed over 200 papers to assess the impact of lockdowns on air quality around the world. They found significant reductions in  $NO_2$  and CO levels, small reductions in  $PM_{2.5}$  and increases in  $O_3$  concentrations. The effects varied by season and region, and the study highlighted the need for future research to include meteorological corrections for accurate results. Only about a third of the studies reviewed included methods for meteorological correction or normalisation.

Matthias et al. (2021) conducted a modelling study for Central Europe, estimating emission reductions for January to June 2020. For secondary pollutants, they found that meteorological effects outweigh the effect by emission reductions from restrictions. Putaud et al. (2023) compared measurements at 28 sites across Europe for spring 2020 with CAMS ensemble forecasts and found a slight decrease in PM<sub>2.5</sub> and PM<sub>10</sub> during the lockdown and a strong increase after the measures were lifted. The study corrects the occurring bias between modelled and measured values by a time-dependent normalisation of the CAMS forecasts to the observations estimated from 2019 data. They concluded that the increased ozone levels due to reduced NO<sub>x</sub> lead to altered oxidation capacities and therefore more SOA formation. The study also analysed data collected in Melpitz and Košetice before, during and after the lockdown in March 2020. In Melpitz, slightly higher PM<sub>2.5</sub> concentrations than expected by CAMS were detected during the lockdown. In May 2020, after the lockdown, they were even twice as high as modelled. In Košetice, the values before and during the lockdown were slightly below the expected values, while the concentrations afterwards were 30% higher. Forster et al. (2020) calculated emission trends based on Google mobility data for six sectors (land transport, residential, energy, industry, public and aviation) per country. These data show that in March 2021, BC emissions from the residential sector in Germany and the Czech Republic were increased by approximately 10%, while BC emissions in all sectors combined were decreased by about 20% compared to a baseline scenario. Mbengue et al. (2023) conducted an extended study analysing the effects of the COVID-19 lockdowns at Košetice using normalisation techniques to account for meteorological effects. They found that during the winter of the second lockdown (December 2020 - February 2021), dispersion normalised concentrations of EC were reduced by 28%, while OC and SOC concentrations increased by 19% and 51%, respectively. They concluded that this was due to a greater influence of emissions from local domestic activities. Considering that our study sites are background stations with low traffic influence and high contribution of domestic heating emissions, locally increased emissions due to the COVID-19 mitigation measures seem plausible, leading to higher PM<sub>2.5</sub> and probably SOA concentrations than without these measures. These changes are not included in the emissions in the model and may be another source of underestimation in the model.

# 4.2 Anthropogenic secondary organic aerosol

Bergström et al. (2012) found an underestimation of winter organic aerosol in a modelling study focusing on several years in Europe. Their conclusion was that emissions from wood combustion are under-represented in current emission inventories. Previous source apportionment studies have shown that residential heating is a significant contributor to SOA formation. Lhotka et al. (2025) identified a relationship between primary organic aerosol (POA) and oxidised organic aerosol (OOA) source factors associated with residential heating. The high contribution of highly oxidised OA in winter can be attributed to the local influence of biomass burning. In contrast, at Melpitz, coal combustion plays a more prominent role in oxidised OA formation, indicating the impact of long-range transport (Atabakhsh et al., 2023). An intensive tagging study by Bartík et al. (2024) utilized the PSAT module in CAMx, supplementing the CAMS emission inventory with more detailed residential emission data for the Czech Republic and additional intermediate - volatility organic compound (IVOC) emissions from wood combustion. Their findings indicate that VOC and IVOC emissions from the 'Other Combustion' sector represent the largest source of SOA in Central Europe during winter, contributing up to 0.4  $\mu$ g m<sup>-3</sup>. In order to investigate whether a potential underestimation of SOA precursors from domestic heating has contributed to the lower than expected concentrations of OM in our model, we have carried out a sensitivity study.

#### 4.2.1 Sensitivity study

The parameterisation of SOA is influenced by two key variables: the precursor gases emitted and the rate at which SOA is formed from these precursors. Previous studies suggest, that phenol is a significant component of emissions from incomplete combustion processes like wood burning. Phenol is one of the key gaseous precursors responsible for the formation of SOA during biomass burning activities (Hatch et al., 2015; Bruns et al., 2016). Liu et al. (2024) highlight the critical role of nighttime NO<sub>3</sub> oxidation of anthropogenic VOCs from biomass combustion, a process they find is inadequately represented in current atmospheric models. Their results show that increasing both phenol emissions and the associated SOA yield leads to a twofold increase in SOA production via NO<sub>3</sub> oxidation across Europe during winter. In our model, phenol is included in the lumped species CSL (cresol and other aromatics) (see equation R1). NMVOC emissions are delivered by the UBA and CAMS emission inventories (Schneider et al., 2016; Kuenen et al., 2022). The NMVOC emission flux is split into the different relevant model species based on emission profiles created by Theloke and Friedrich (2007) for 306 individual species including phenol. These profiles are based on a NMVOC source database from 1990 (Olivier et al., 1996) and do not include phenol emissions from domestic heating. Therefore, it is reasonable to assume that the VOCs emitted from domestic heating are not fully captured by the model. Natural fire emissions provided by GFAS do not include CSL, but toluene (TOL) and xylene (XYL) emissions. Given the low impact of natural fires in winter in Europe, it can be assumed that they do not contribute much to the formation of secondary particles.

$$CSL + OH \rightarrow \alpha_1 CVARO1 + \alpha_2 CVARO2$$

$$CSL + NO_3 \rightarrow \alpha_1 CVARO1 + \alpha_2 CVARO2$$
(R1)

The SORGAM module (see section 2.3) estimates SOA formation from aromatic precursors using data from Odum et al. (1997), who conducted smog chamber experiments to quantify SOA production from gasoline vapor. This method is therefore primarily tailored to traffic emissions. However, for aromatic precursors emitted by domestic heating, an increase in SOA yield aligned with phenol SOA formation rates is more suitable. Due to the limited availability of chamber studies on phenol gasphase SOA formation, we derived a new yield estimate based on four OH oxidation measurements from Yee et al. (2013). Given the importance of nocturnal oxidation, we also applied these modifications to the NO<sub>3</sub> reaction. A non-linear least squares fit for the  $\alpha_i$  values was performed with fixed  $K_{om,i}$  coefficients ( $K_{om,1}$ = 0.2899,  $K_{om,2}$ = 0.0103). As  $\alpha_2$  yielded negative values, we decided to keep  $\alpha_2$  fixed and performed the fit solely for  $\alpha_1$  (see Fig. A6 in the Appendix). These adjustments result in an approximately threefold increase in SOA yield.

Further, an adjustment of the input emission was done. To get a good representation of phenol emissions from domestic heating processes, we decided to scale the emissions to the CO emissions of the emission sector 'other combustion'. Following wood combustion chamber studies from Bruns et al. (2016) on average all NMVOC emissions make up 0.22 times the CO emissions. According to Schauer et al. (2001), phenol and substituted organic compounds are approximately 10% of the overall NMVOC emissions from wood combustion. Accordingly, we set our 'other Combustion' sector CSL emissions to 0.022 times the sector's CO emissions.

We simulated three sensitivity runs to compare these adjustments. First with the adjusted SOA yield alone (S1), second with the new CSL emissions alone (S2), and third with both combined (S3). Table 4 gives an overview of the coefficients and emissions used in the different sensitivity runs and the original base run. All sensitivity runs were performed for our middle domain, TRACED0 (see Fig. 1), as it provides the best trade-off between spatial resolution and area coverage. The three sensitivity runs were not nested, but use the same initial and boundary conditions as the base run.

**Table 4.** Overview of the sensitivity simulations. Shown are changes to  $\alpha_1$  to adjust the SOA yield parameter for aromatic precursors and scaling of CSL emissions based on CO emissions from GNFR C to account for phenol contributions.

| simulation | stoichiometric coefficient |          | CSL emissions                      |
|------------|----------------------------|----------|------------------------------------|
|            | $\alpha_1$                 | $lpha_2$ | 'other Combustion'                 |
| base run   | 0.039                      | 0.108    | CAMS NMVOC split                   |
| S1         | 0.219                      | 0.108    | CAMS NMVOC split                   |
| S2         | 0.039                      | 0.108    | $0.022 \times \text{CO}$ emissions |
| <b>S</b> 3 | 0.219                      | 0.108    | $0.022 \times \text{CO}$ emissions |

# 4.2.2 Sensitivity study results




The changes in CSL emission flux and the corresponding mean OM concentration across the three sensitivity runs are presented in Table 5. The values for each station represent the model result from the 4×4 km grid cell in which the station is located. In scenario S1, the increased SOA yield for aromatic precursors has the most pronounced effect in urban areas, as it influences emissions from all source sectors, including industry and transport. The adjusted SOA yield applies to both daytime OH

Table 5. Changes in CSL emission flux and mean organic matter in PM<sub>2.5</sub> for all sensitivity runs compared to the base run.

|          | CSL emission [μg m <sup>-2</sup> s <sup>-1</sup> ] | OM mean [μg m <sup>3</sup> ] |              |              |  |  |  |
|----------|----------------------------------------------------|------------------------------|--------------|--------------|--|--|--|
|          | S2, S3                                             | S1                           | S2           | S3           |  |  |  |
| domain   | + 0.00001 (0.08%)                                  | + 0.27 (18%)                 | + 0.07 (4%)  | + 0.62 (39%) |  |  |  |
| Melpitz  | - 0.0016 (- 60%)                                   | + 0.52 (39%)                 | + 0.21 (16%) | + 0.77 (58%) |  |  |  |
| Košetice | + 0.0115 (202%)                                    | + 0.21 (13%)                 | + 0.12 (7%)  | + 0.67 (40%) |  |  |  |
| Frýdlant | + 0.0230 (188%)                                    | + 0.42 (23%)                 | + 0.28 (15%) | + 0.95 (53%) |  |  |  |

oxidation and nighttime NO<sub>3</sub> oxidation. Among the stations, Melpitz shows the highest relative increase (39%) in mean OM concentrations, reaching 1.86  $\mu$ g m<sup>-3</sup>, due to high aromatic precursor levels.

In scenario S2, the emissions of aromatics from domestic heating are introduced as CSL emissions by the sector 'other Combustion'. Although total CSL emissions across the domain remain constant, their spatial distribution shifts: emissions decrease in Melpitz but increase significantly in Frýdlant and Košetice. The domain-wide mean OM concentration shows an overall modest increase of 4%, with the largest increases observed in the central Czech Republic and southern Poland, where domestic heating sources are abundant. Interestingly, despite a reduction in CSL emissions at Melpitz compared to the base run, OM concentrations at Melpitz increase similarly to those at Frýdlant (+16% at Melpitz and +15% at Frýdlant). This is attributed to increased CSL emissions in the surrounding areas and the transport of SOA and its precursors to the site. These findings align with previous studies: Poulain et al. (2011) linked winter OM at Melpitz to transported particles, while Spindler et al. (2012) reported that SOA concentrations peaked in winter air masses arriving from the east, highlighting the role of anthropogenic precursor-driven SOA formation during long-range transport. This is also consistent with the conclusions of Atabakhsh et al. (2023).






In the combined S3 run, average OM concentrations in Melpitz increase by 58% to  $2.11~\mu g\,m^{-3}$ , representing the highest relative impact among all stations. This increase can be attributed to enhanced SOA transport and formation from aromatic precursors. Frýdlant shows the largest absolute OM increase, with an average increment of  $0.95~\mu g\,m^{-3}$ , reaching  $2.76~\mu g\,m^{-3}$  (see Table 5). Figure 9 compares diurnal OM cycles from the base and sensitivity runs with measurements. At Frýdlant and Košetice, the combined adjustments in S3 produce greater impacts on OM concentrations compared to the individual sensitivity runs. This leads to a better agreement with the measurements in Košetice but results in overestimation compared to the AMS/ACSM data at Frýdlant. As discussed previously, the discrepancies between the AMS/ACSM and Sunset measurements cannot be fully resolved in this work, and both datasets must therefore be regarded as valid. Taking into account the measurement uncertainties, the fact that the simulated OM concentrations at Frýdlant now lie between the two measurements supports the plausibility of the modelled increase. Evaluating both datasets in combination provides a more comprehensive and balanced assessment of actual OM levels. The AMS/ACSM is better suited to capture diurnal patterns due to its higher time resolution. At Frýdlant, the model simulates a clear morning peak in OM concentrations that is absent in the AMS/ACSM data. This discrepancy suggests that the model may be overestimating the contribution from local or near-field sources while underestimating the influence of long-range transport.

At Melpitz, however, the difference between S1 and S3 is less distinct, suggesting that higher baseline precursor concentrations already contribute significantly to SOA formation at this site. The correlation of the modelled SOA with AMS/ACSM data improves in Košetice, with the correlation coefficient increasing from 0.18 to 0.29, while Melpitz and Frýdlant show no significant improvement. Although the model now better reflects SOA contributions at Košetice, overall OM concentrations remain underestimated.


Figure 9. Hourly diurnal cycle of organic matter in  $PM_{2.5}$  over the entire measurement period for all sensitivity runs compared to the base run. Bar graphs show averaged filter data based on 12-hour sampling intervals, with whiskers representing a measurement uncertainty of  $\pm 12\%$  (Karanasiou et al., 2020). AMS/ACSM data are shown with a measurement uncertainty of  $\pm 25\%$  (Canagaratna et al., 2007).

Figure 10. Difference between the averaged modelled organic matter in PM<sub>2.5</sub> in the S3 and base simulation.

The spatial pattern of the increase in OM concentrations in S3 compared to the base run is shown in Fig. 10. The strongest increases occur in the Czech Republic and southern Poland. In particular, the city of Prague and its outskirts show a high increase, which is consistent with the small residential houses in the suburbs and surrounding areas still using coal and wood combustion for heating (Schwarz et al., 2008; Makeš et al., 2021). Domestic heating emissions likely increased during the COVID-19 measures as more people stayed at home, contributing to higher PM<sub>2.5</sub> and possibly SOA concentrations (Mbengue et al., 2023). Additionally, the campaign coincided with an unusually cold period in early February, which likely increased heating activity further. This effect was not fully captured by the model.

The simulation of the OM peak in early February, after the snow event, shows no noticeable improvement with the sensitivity runs, likely linked to the overall underestimation of emissions (see Fig. A8 in the appendix). Measurements reveal a distinct OM concentration peak on 3 March, particularly at Košetice and Frýdlant. The S3 run captures the peak reasonably well at Frýdlant but still underestimates OM at Košetice. HYSPLIT backward trajectories for Frýdlant on this date indicate a significant influence from air masses passing over the Czech Republic and Košetice (see Fig. A7 in the Appendix). At Košetice, wind direction shifts from east to west, as simulated by the model, while at Frýdlant, wind direction remains steady, allowing precursor accumulation and increased SOA formation.

Overall, the sensitivity studies showed that scaling AVOC emissions from wood combustion to residential heating emissions improves the spatial distribution of SOA in the study area. Long-range transport of precursors and SOA is captured as well as the local influence on OM concentrations.

### 5 Conclusion, Limitations and Future Directions




The study investigates the sources of primary and secondary anthropogenic organic aerosol in Central Europe during winter. The chemical transport model COSMO-MUSCAT was used to analyse concentrations of particulate matter, in particular particles originating from combustion processes. The model results were compared with measurements made in Germany and the Czech Republic in terms of overall PM<sub>2.5</sub> concentration and concentrations of individual species. The model underestimated the total PM<sub>2.5</sub>, especially during high concentration peaks. A pronounced underestimation occurred in early February, likely

due to the prevailing meteorological conditions combined with changed heating behaviour. During this period, all tracers were underestimated, whereas after early February, the model accurately captured the behaviour of most tracers. However, the underestimation of PM<sub>2.5</sub> during concentration peaks remained evident. The discrepancies in modelled PM<sub>2.5</sub> concentrations do not appear to be due to deviations in EC, sulfate, or nitrate levels, but rather to the underestimation of OM. Although the present study reproduced diurnal OM profiles well at two monitoring sites, measurements at Košetice are underestimated, partly due to an inadequate representation of SOA formation from residential heating (wood combustion), a major source of anthropogenic VOCs. These AVOCs contribute considerably to the formation of SOA, and it is likely that their insufficient representation in our model contributes to the overall underestimation of OM during winter. The effect is most pronounced in the central Czech Republic, where the basin-like topography allows air masses to linger, promoting the accumulation of emissions and extended SOA formation. We found a higher contribution of domestic heating in the eastern part of our study region, which is accompanied by high concentrations of OM, especially at the station in Košetice. Sensitivity tests with adjustment for SOA yields and AVOC emissions showed an average increase in OM concentrations of over 40% at the measurement sites. It is likely that the model underestimates SOA precursor emissions from domestic heating sources, as well as from additional sources that are missing or unaccounted for in the underlying emission inventory. A more detailed inventory, as used in Bartík et al. (2024) for the Czech Republic, reveals a redistribution of total primary PM<sub>2.5</sub> residential combustion emissions from urban to rural areas, compared to the inventory used in this study. In addition, the model may underestimate the contribution of SOA precursors other than phenol. Implementing more detailed and up-to-date emission inventories that provide information on the types of fuels used, their spatial distribution, and their temporal profiles offers strong potential to enhance the model's overall performance for OM. Our findings highlight that regional domestic heating emissions contribute significantly to overall air pollution in the study area. Addressing these emissions is complex, as they are hard to quantify and regulations for private households are more challenging to implement. Consequently, obtaining more detailed information on these sources is vital for developing targeted and feasible measures. Besides updated time profiles representing seasonal, weekly and daily patterns of emissions, changing heating behaviour due to extreme meteorological conditions could be taken into account by implementing a temperature dependence of emissions. The Heating Degree Day (HDD) approach, introduced by Guevara et al. (2021), considers the influence of outdoor temperature on heating activity and its associated emissions. A recent study by Guion et al. (2024) enhanced this method by incorporating country-specific and species-specific parameters, demonstrating improved temporal correlations and more accurate detection of PM emission threshold exceedances compared to simulations using fixed parameters or monthly temporal factors. Implementing this approach in COSMO-MUSCAT could enhance the accuracy of our model results during winter. In addition, heating emissions may not only be underestimated in quantity, but the contribution of different fuel types to the domestic heating sector may also vary with temperature, as additional coal burning in households may occur during colder periods.








Comparing the non-reactive tagging approach in COSMO-MUSCAT to measurement-based, receptor-oriented source apportionments can further evaluate its capability and identify areas for improvement. This comparison can provide valuable insights into the performance of the model and guide future refinements. By addressing these gaps and incorporating the necessary updates, such as updated emission inventories, improved SOA yields and model evaluation through comparison with

measurement data, the model could provide a more comprehensive and accurate representation of SOA formation processes, 630 enabling better understanding for air quality management. Data availability. Modeling data supporting the findings of this article are available online: DOI:10.5281/zenodo.16406515. Measurement data will be made available upon request. Please contact H. Herrmann (herrmann@tropos.de)

# Appendix A

**Figure A1.** Time profile applied to the GNFR C emission factor following Kuenen et al. (2014)

**Table A1.** Splitting factors applied in this study to disaggregate CAMS PM2.5 and PM10 bulk emissions into individual subgroups within the GNFR C emission sector (Kuenen et al., 2022).

|               | Czech Republic | Germany     | Poland      |
|---------------|----------------|-------------|-------------|
| PM2.5         |                |             |             |
| EC_fine       | 0.504027533    | 0.439516267 | 0.575485492 |
| OM_fine       | 0.424751227    | 0.427286603 | 0.271872046 |
| SO4_fine      | 4.61002E-05    | 0.00084613  | 3.27417E-05 |
| Na_fine       | 1.00147E-05    | 0.000184478 | 4.68024E-06 |
| OthMin_fine   | 0.071165125    | 0.132166522 | 0.152605039 |
| PM10          |                |             |             |
| EC_coarse     | 0.145269867    | 0.177740784 | 0.088907128 |
| OM_coarse     | 0.000813148    | 0.000401242 | 0.001754957 |
| SO4_coarse    | 0              | 0           | 0           |
| Na_coarse     | 0.006387868    | 0.008082688 | 0.001165384 |
| OthMin_coarse | 0.847529117    | 0.813775287 | 0.908172531 |

**Table A2.** Splitting factors applied in this study to disaggregate CAMS NMVOC bulk emissions into individual subgroups within the GNFR C emission sector (Kuenen et al., 2022).

|                                | Czech Republic | Germany     | Poland      |
|--------------------------------|----------------|-------------|-------------|
| alcohols                       | 0.112890083    | 0.116886184 | 0.094423891 |
| ethane                         | 0.062493957    | 0.057406665 | 0.077011717 |
| propane                        | 0.020926633    | 0.018650187 | 0.033919437 |
| butanes                        | 0.008632808    | 0.009624955 | 0.015227517 |
| pentanes                       | 0.012577632    | 0.017491698 | 0.010032825 |
| hexanes and higher alkanes     | 0.009691263    | 0.011952296 | 0.008146194 |
| ethene                         | 0.122201269    | 0.113657698 | 0.151810489 |
| propene                        | 0.053424416    | 0.051930621 | 0.052867337 |
| ethyne                         | 0.043099331    | 0.042365695 | 0.044307905 |
| monoterpenes                   | 0              | 0           | 0           |
| other alk(adi)enes and alkynes | 0.058142299    | 0.057501879 | 0.056860258 |
| benzene                        | 0.067178835    | 0.067837987 | 0.06428175  |
| toluene                        | 0.029062597    | 0.02989825  | 0.025864856 |
| xylene                         | 0.00899594     | 0.008851544 | 0.00917539  |
| trimethylbenzenes              | 3.39124E-06    | 0.00016824  | 1.40621E-05 |
| other aromatics                | 0.00658469     | 0.006790321 | 0.005505717 |
| esters                         | 0              | 0           | 0           |
| ethers                         | 0.047009274    | 0.047300576 | 0.039226104 |
| chlorinated HC's               | 0              | 0           | 0           |
| methanal                       | 0.019953442    | 0.022564238 | 0.016412027 |
| other alkanals                 | 0.060239696    | 0.063909539 | 0.050490655 |
| ketones                        | 0.007555396    | 0.009250493 | 0.006416798 |
| acids                          | 0.249303136    | 0.244278533 | 0.237864452 |
| others                         | 3.39124E-05    | 0.001682401 | 0.000140621 |

**Table A3.** Overview of all the tagged combinations of variable, source region and source sector.

| Variable                           | Region         | Sector                        |
|------------------------------------|----------------|-------------------------------|
| OMfine, ECfine, OMcoarse, ECcoarse | Germany        | GNFR A - Public Power         |
|                                    |                | GNFR B - Industry             |
|                                    |                | GNFR C - Other Combustion     |
|                                    |                | GNFR F1- Traffic: Gasoline    |
|                                    |                | GNFR F2 - Traffic: Diesel     |
|                                    | Poland         | GNFR A - Public Power         |
|                                    |                | GNFR B - Industry             |
|                                    |                | GNFR C - Other Combustion     |
|                                    |                | GNFR F1- Traffic: Gasoline    |
|                                    |                | GNFR F2 - Traffic: Diesel     |
|                                    | Czech Republic | GNFR A - Public Power         |
|                                    |                | GNFR B - Industry             |
|                                    |                | GNFR C - Other Combustion     |
|                                    |                | GNFR F1- Traffic: Gasoline    |
|                                    |                | GNFR F2 - Traffic: Diesel     |
|                                    | Boundary       | GNFR A - Public Power         |
|                                    |                | GNFR B - Industry             |
|                                    |                | GNFR C - Other Combustion     |
|                                    |                | GNFR F1- Traffic: Gasoline    |
|                                    |                | GNFR F2 - Traffic: Diesel     |
|                                    | Total          | GNFR A - Public Power         |
|                                    |                | GNFR B - Industry             |
|                                    |                | GNFR C - Other Combustion     |
|                                    |                | GNFR D - Fugitives            |
|                                    |                | GNFR F1 - Traffic: Gasoline   |
|                                    |                | GNFR F2 - Traffic: Diesel     |
|                                    |                | GNFR F4 - Traffic: Non-Exhaus |
|                                    |                | GNFR I - Off Road             |
|                                    |                | GNFR K - Livestock            |
|                                    |                | GNFR L - Agriculture          |
|                                    |                | Other                         |

**Figure A2.** Left: Time series for organic matter mass concentration for the three sites. Comparison of measured AMS/ACSM data with the modelled primary OM concentration, split into the share with origin outside the N0 domain, origin inside the study domains and secondary organic aerosol. Right: normalised POA to total OM ratio

Figure A3. Time series for elemental carbon concentration for the three sites. Comparison of Aethalometer and Sunset Filter data and modelled data.

Figure A4. Time series for nitrate mass concentration for the three sites. Comparison of measured and modelled data.

Figure A5. Time series for sulfate mass concentration for the three sites. Comparison of measured and modelled data.

Figure A6. Non-linear least squares fit for two-product aromatic class.

**Figure A7.** Backward trajectory ending 03 March 2021 at 12, 15 and 18 UTC in Frýdlant, created with NOAA HYSPLIT Trajectory Model (Rolph et al., 2017; Stein et al., 2015)

Figure A8. Time series for organic matter mass concentration for all sensitivity runs compared to the base run and measurements.

**Table A4.** Relative contributions of different source sectors and source regions to the total of ECfine, ECcoarse, OMfine and OMcoarse and absolute mean over all sectors. Contributions from outside the N0 domain are not included.

|               |                       | ECfine [9 | 6]       | ECcoarse [%] |         |          | OMfine [%] |         |          | OMcoarse [%] |         |          |          |
|---------------|-----------------------|-----------|----------|--------------|---------|----------|------------|---------|----------|--------------|---------|----------|----------|
|               |                       | Melpitz   | Košetice | Frýdlant     | Melpitz | Košetice | Frýdlant   | Melpitz | Košetice | Frýdlant     | Melpitz | Košetice | Frýdlant |
| sector        | Public Power          | 0.6       | 0.3      | 0.9          | 0.9     | 2.9      | 5.3        | 0.5     | 0.3      | 0.7          | 0.1     | 0.2      | 0.8      |
|               | Industry              | 2.0       | 0.5      | 0.8          | 14.0    | 7.5      | 7.2        | 2.4     | 0.5      | 1.1          | 7.3     | 2.1      | 5.8      |
|               | other Combustion      | 34.6      | 72.9     | 76.3         | 5.5     | 18.8     | 30.9       | 30.9    | 69.6     | 72.6         | 0.0     | 0.1      | 0.4      |
|               | Traffic: gasoline     | 0.9       | 0.1      | 0.1          | 0.0     | 0.0      | 0.0        | 1.6     | 0.2      | 0.4          | 0.0     | 0.0      | 0.0      |
|               | Traffic: diesel       | 10.1      | 3.0      | 4.6          | 0.5     | 0.1      | 0.1        | 3.6     | 1.3      | 2.1          | 0.0     | 0.0      | 0.0      |
|               | Fugitives             | 1.2       | 0.3      | 0.5          | 27.1    | 17.4     | 25.5       | 0.4     | 0.1      | 0.2          | 2.4     | 1.6      | 2.8      |
|               | Traffic: non-exhaust  | 0.5       | 0.1      | 0.1          | 7.9     | 7.3      | 6.6        | 1.8     | 0.4      | 0.6          | 8.8     | 7.8      | 14.1     |
|               | off road              | 14.4      | 3.1      | 2.2          | 24.9    | 8.8      | 5.5        | 15.4    | 4.8      | 3.6          | 0.0     | 0.0      | 0.0      |
|               | Livestock             | 0.0       | 0.0      | 0.0          | 0.0     | 0.0      | 0.0        | 4.1     | 1.2      | 1.1          | 40.2    | 37.0     | 34.5     |
|               | Agriculture           | 0.0       | 0.2      | 0.1          | 0.1     | 5.0      | 1.1        | 0.3     | 0.7      | 0.2          | 16.2    | 18.0     | 12.4     |
|               | other                 | 2.1       | 0.3      | 0.6          | 1.2     | 0.4      | 0.5        | 5.1     | 1.0      | 2.0          | 0.0     | 0.0      | 0.0      |
|               | Boundary              | 33.2      | 19.1     | 13.6         | 17.5    | 30.7     | 16.8       | 33.5    | 19.8     | 15.2         | 20.2    | 28.4     | 26.0     |
| country       | Czech Republic        | 6.9       | 76.8     | 63.0         | 7.0     | 49.8     | 34.8       | 6.1     | 75.8     | 63.3         | 2.0     | 55.8     | 32.6     |
|               | Germany               | 55.5      | 2.5      | 7.8          | 71.9    | 13.2     | 24.0       | 56.6    | 3.0      | 9.0          | 70.2    | 8.9      | 22.1     |
|               | Poland                | 4.2       | 1.4      | 15.2         | 3.3     | 5.1      | 23.8       | 3.5     | 1.0      | 12.0         | 3.0     | 2.1      | 16.0     |
| absolute mean | [μg m <sup>-3</sup> ] | 0.3496    | 0.9838   | 1.0504       | 0.0775  | 0.0422   | 0.0611     | 0.3432  | 0.8624   | 0.8649       | 0.1094  | 0.0659   | 0.0579   |

Competing interests. The authors declare that they have no conflict of interest.

Author contributions. Conceptualization: HW

Data curation: HW Formal analysis: HW

Investigation: HW, SA, LP, RL, PV, JS, PP, JO

Methodology: RW, RS, ML, HW

Software: RW, HW

Supervision: RW, RS, LP

Visualization: HW

Writing: original draft preparation: HW

Writing: review & editing: HW, RS, RW, LP, RL, PV, VZ, IT

Funding acquisition: VZ, HH, IT

Acknowledgements. We thank Ondřej Vlček from the Czech Hydrometeorological Institute for providing the Czech emission data that we used in Bartík et al. (2024). The authors acknowledge the use of DeepL Write [https://www.deepl.com/write] and ChatGPT [https://chat.openai.com/] to identify improvements in the writing style.

*Financial support.* This work was supported by the GACR under grant 20-08304J and by the DFG under grant 431895563 as part of the German-Czech cooperation within the project TRACE.

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
