# Peer review of "Modeling Anthropogenic Aerosol Sources and Secondary Organic Aerosol Formation: A Wintertime Study in Central Europe"

_EGUsphere, 2025_

## Referee Comment (RC2)

The manuscript details the use of a non-reactive tagging method of tracking primary organic matter (OM) using the model COSMO-MUSCAT to investigate the contribution of residential heating to OM during the winter of 2021, at 3 measurement stations across central Europe. The findings indicate that the modelled OM is underestimated at these sites, which is mainly attributed to the current under-representation of wood combustion SOA. The study is well framed, and conclusions are adequately presented. I recommend the publication of this work, before the authors clarify and accommodate the following questions/recommendations.

L6:8: "Although the magnitude and temporal changes of the model results mostly agree with total OM values at two measuring stations, it appears to underestimate measurements at a site in the central Czech Republic." It would be better to mention earlier in the introduction that there was 1 measurement site in Germany and 2 in CR to avoid confusion.

L18: " wider societal costs associated with it". interesting point but would benefit from stating the wider societal costs.  The values can be presented here to underline the magnitude of the costs and losses.

L71: "The second main wind direction is East (about 17% of the time), with dry continental air masses influenced by long-distance
transport from Poland, Belarus, Ukraine, Slovakia and the Czech Republic (Spindler et al., 2001, 2012, 2013).". Is this during summer or winter?

L135: Add the values of splitting profiles in the appendix.

L163: wouldn't it benefit to use a 2-d VBS method over the 2 product Odum parameterization?

L196: add a table in the appendix detailing the main tagged species

L208-209: Since gas phase species and aerosol chemistry is not considered, can the authors quantify how much the SOA will be under-predicted based on the lack of these processes in the model.

L249: is there a comparison of simulated and measured boundary layer height?

L253-254: provide value for the "slight difference".

Figure 4: simulated PM2.5 and AMS PM1 are not directly comparable. for e.g if we consider Nitrate how can one interpret high model PM2.5 conc and low AMS PM1 concentrations. Both the values could be in similar if we consider only modelled PM1. Also, 4d is comparing PM1, PM2.5 and PM10. Are the PM1, pm2.5 and PM10 masses correlated?  It would perhaps be better to compare, relative or normalized nitrate, sulphate, OM and EC concentrations if one must compare PM1, PM2.5 and PM10 conc.

L297: "The model underestimates the OM concentrations in Košetice (RMSE: 6.48 µg m−3) while for Melpitz and Frýdlant the overall fit is good (RMSE: 1.17 and 2.01 µg m−3)." doesn't this negate the earlier claim that the OM is underestimated in the simulations?

L310-315: Since the winter values are heavily meteorological dependent one must mention if the weather patterns during the said years matched 2021. From the description above the year 2021 seems to be an odd one considering the long Sahara dust events and the cold periods. I would suggest making such comparisons to more tangible SOA concentrations, which can then present a broader picture of a trend in SOA underestimation.

L 345: Public power contribution. This is interesting. One would expect higher contributions from public power at Kosetice especially in the cold period as the air masses is stagnant. Can you explain why is the contribution of public power low during the cold period?

L 358-359: Melpitz cross border transport.
Is this during the cold or warm period? Looking at Figure 7, it appears that Kosetice and Frydlant have larger cross-border (Poland) contribution to public power than Melpitz. Also, it appears that at Melpitz the cross border contibutions is more in the warm period but at kosetic the polish and german contributions are significant even during the cold period.

Figure 7: I would suggest removing the wind barbs since it doesn't add any information's. or did the authors miss the y axis with the degrees?

Table 3: why is alpha 1 same for S1 and S3? Shouldn't it be same for S2 and S3? Some explanation is needed in the main body or the table caption.

L496-497: it would be interesting to see if the increase in OM holds true during the cold period of stagnant airmass at these sites.

L 537-538: I would suggest saying that the diurnal profiles are reproduced, not the total OM magnitude.

---

## Author Comment (AC1)

**Anonymous Referee #1, 02 Jun 2025**

Reviewer comment: *'Manuscript egusphere-2025-1225 by H. Wiedenhaus et al. reports model results of the COSMO-MUSCAT chemistry transport model on particulate matter concentrations during winter months, with a focus on emissions from residential heating. Modeled organic matter concentrations are compared against measurements at three observation sites with specific aerosol instrumentation. The employed source apportionment by tagging method is robust and carefully applied (only for primary components). The study investigates the impact of SOA formation from anthropogenic VOCs related to wood burning emissions. A weak component of the model system is the emission inventory for residential heating. The study should try to better identify and isolate wood burning as the missing source of primary/secondary OM. The sensitivity tests are well performed but it is difficult to evaluate the impact on anthropogenic SOA concentrations and their spatial distribution. I strongly recommend the addition of one more sensitivity test including more detailed wood burning emissions. The conclusions are based on the findings of the model study and future directions are well formulated.'*

**Author's response:**

We thank the reviewer for the thorough review and suggestions. We have answered all comments below and outlined changes to the manuscript text. It is true that wood combustion emission inventories are relatively uncertain both in contribution to EC and OM generally as well as in spatial distribution. As shown in our study additional SOA precursor species can enhance OM while maintaining EC. As outlined later, a different spatial distribution as provided in the emission inventory by Bartik et al., 2024, would enhance particulate matter from residential combustion in rural regions while decreasing it in urban centers (data only available for Czech Republic). We have included a section discussing these shortcomings of the current emission data set in the manuscript. However, running another simulation with combined emission data sets is beyond the scope of the study but will be considered for a follow up study.
* * *
**Reviewer Comment:**

1.) Introduction (P2, line 28-34): Suggest rewriting the paragraph on transboundary transport of pollution to Germany. Expand on the influence of long-range transport from eastern Europe, also including reports from EMEP. The sentence in line 28 ("the inflow of air masses from the east") is not logical and should be deleted.

**Author's response:**

The sentence was removed as suggested and the paragraph was changed to improve its understandability.

**Author's changes in manuscript:**

Line 30 ff:

'In this transition zone between less and more polluted regions, the rural background station Melpitz in eastern Germany recorded the highest annual mean $PM_{10}$ concentration in 2021 as reported by the 'European Monitoring and Evaluation Programme' (EMEP) (Fagerli et al., 2023). Previous studies in Germany have shown that long-range transport from Eastern Europe, particularly from combustion processes, is a major contributor to regional background particle concentrations (van Pinxteren et al., 2019, 2016). The inflow of air masses from the east was associated with $PM_{10}$ concentration peaks leading to an increase in exceedances of the current daily limit value of 50 µg m$^{-3}$ (van Pinxteren et al., 2019). However, the relative contributions of multiple combustion sources to primary and secondary paticles, as well as their transboundary transport remain insufficiently quantified. This needs to be better characterised to enable effective and better targeted mitigation strategies to address the prevailing air quality challenges.'
* * *
**Reviewer Comment:**

2.) Introduction (P2, line 54-58): Add a brief description of the TRACE project, its objectives and how this study addresses the project's objectives.

**Author's response:**

We have included a brief overview of the TRACE project and its objectives.

**Author's changes in manuscript:**

Line 58 ff:

'The TRACE project: 'Transport and Transformation of Atmospheric Aerosol over Central Europe with an Emphasis on Anthropogenic Sources', aims to develop a comprehensive understanding of the contribution of transported anthropogenic aerosols relative to local emissions, integrating expertise in synergistic measurement methods and modelling tools.'
* * *
**Reviewer Comment:**

3.) Model description: how frequent is the exchange of variables between the meteorological and the chemistry-transport component? What is the expected advantage of the online coupling specifically for this study compared to using an offline coupled CTM?

**Author's response:**

COSMO and MUSCAT work widely independent on different grid structures and have their own time step control. The coupling procedure is adapted to the applied IMEX scheme for the numerical solution of the three-dimensional advection-diffusion-reaction equations in the chemistry-transport code MUSCAT (Lieber and Wolke, 2008; COSMO-MUSCAT description). This IMEX scheme uses explicit second order Runge-Kutta methods for the integration of the horizontal advection and an implicit method for other processes such as chemical reactions (Wolke and Knoth, 2000; Schlegel et al., 2012a, b). The fluxes resulting from the horizontal advection are defined as a linear combination of the fluxes from the current and previous stages of the Runge-Kutta method. These horizontal fluxes are treated as ''artificial'' sources within the implicit integration. A change of the solution values as in conventional operator splitting is thus avoided. Within the implicit integration, the stiff chemistry and all vertical transport processes (turbulent diffusion, advection, deposition) are integrated in a coupled manner by the second order BDF (Backward Differentiation Formula) method.

Coupling between meteorology and chemistry-transport takes place at each horizontal advection time step (15 - 80 seconds or lower, if necessary, due to the CFL criterium). All meteorological fields are given with respect to the uniform horizontal meteorological grid. They have to be averaged or interpolated from the base grid into the block-structured chemistry-transport grid with different resolutions. The coupling scheme provides time-interpolated meteorological fields (vertical exchange coefficient, temperature, humidity, density) and time-averaged mass fluxes. The coupling scheme allows the highly time-resolved forcing of the chemistry-transport calculations by the meteorological model (each time step).  The advantage over offline coupled model calculations lies, on the one hand, in the higher temporal resolution of the meteorological input data (e.g., wind fields, temperature, humidity, turbulent exchange coefficients) and, on the other hand, in the consistent description of transport processes (e.g., deposition, mixing layer height, and vertical exchange).

*Lieber, M. & Wolke, R. (2008). Optimizing the coupling in parallel air quality model systems. Environ. Model. Softw., 23, 235-243.*

*Wolke, R., and Knoth, O., Implicit-explicit Runge-Kutta methods applied to atmospheric chemistry-transport modelling, Environ. Model. Softw., 15, 711–719, 2000.*

*Schlegel, M., Knoth, O., Arnold, M. & Wolke, R. (2012a). Implementation of multirate time integration methods for air pollution modelling. Geosci. Model Dev., 5, 1395–1405. doi.org/10.5194/gmd-5-1395-2012.*

*Schlegel, M., O. Knoth, M. Arnold, & R. Wolke (2012b). Numerical solution of multiscale problems in atmospheric modeling. Appl. Numer. Math., 62(10), 1531-1543. doi:10.1016/j.apnum.2012.06.023.*

**Author's changes in manuscript:**

Line 133 ff:

'COSMO and MUSCAT operate largely independently on separate grids and are coupled at each horizontal advection time step (every 15–80 seconds), allowing highly time-resolved meteorological input for the chemistry-transport model.'

**Reviewer Comment:**

4.) Model description: it is stated that the GRETA emission database was provided with resolution of 0.5 km x 1 km. The usual GRETA grid has a resolution of 1 km x 1km. Why did you choose this resolution and was any reprojection on the COSMO grid required? It would be good to mention the specific temporal profile for other combustion (i.e., for residential heating).

**Author's response:**

Thank you for pointing this out. The GRETA emission data were provided by Umweltbundesamt (UBA) at a resolution of 1 km × 1 km. In order to combine it with CAMS data, this was remapped to 0.01° x 0.01° (~0.5 km x 1 km). We have corrected the manuscript to reflect that the overall resolution of the original GRETA emission data used in our model is 1 km × 1 km. It is a good point to include the temporal emission profiles for the 'Other Combustion' source sector. We added the minimum and maximum weighting factor for GNFR C in the text and the entire time profile for this sector in the appendix.

**Author's changes in manuscript:**

Line 135 ff:

'Emissions within Germany are provided by the GRETA database of the German Federal Environment Agency (UBA) (Schneider et al.,2016) for the year 2019 (resolution: 1 km x 1 km). For European emissions outside Germany the CAMS-REG-v5 emission inventory for the year 2018 (resolution: 6 km x 6 km) is used, provided by the Copernicus Atmosphere Monitoring Service (CAMS) (Kuenen et al., 2022).'
* * *
**Reviewer Comment:**

5.) Emissions: Different years of the emissions were used as emissions for 2021. Was something done to adjust for year-to-year changes in emissions? How is the expected variability between the years for the different source types?

**Author's response:**

No, the emission inventory used in this study has not been adjusted to reflect year-to-year changes. We do not anticipate substantial shifts in the underlying emissions for most sectors during the study period, with the exception of the potential effects of COVID-19-related restrictions. These influences are acknowledged and discussed in the manuscript, but they were not explicitly implemented in the emission input.

We also recognize that heating-related emissions, particularly in GNFR sector C ("Other Combustion"), may vary due to interannual differences in ambient temperature. Introducing a temperature-dependent emission factor for this sector would allow for a

more accurate representation of these variations. However, this refinement is planned for future work and is beyond the scope of the present study. A comparison of the recent years 2014-2018 of CAMS GNFR-C emissions shows a deviation of max. +/- 2 % from the mean over these years for our study region (TRACE D1 domain). Therefore, the uncertainties introduced by differences in annual emission of the emission inventory between different recent years are likely smaller than a more realistic temperature-dependent day-to-day variability.

**Author's changes in manuscript:**

None
* * *
**Reviewer Comment:**

6.) SOA formation (P 6, line 160-170): A table should be added with a list of the different model surrogates of SOA precursors from the different parent VOCs. In particular, the precursors of anthropogenic SOA should be detailed. If possible, supplement the relevant reactions and stoichiometric yields of the two pseudo-products.

**Author's response:**

We have added to the manuscript that the detailed information on SOA classes, their reactions, and stoichiometric coefficients can be found in the supplement of Luttkus et al. (2022).

To improve understanding, we have included the reaction equations of the CSL oxidation in the description of the sensitivity study.

**Author's changes in manuscript:**

Line 185 ff:

'All information regarding the precursor VOCs, SOA class names in both the gas and particle phases, along with the reactions and stoichiometric coefficients can be found in Schell et al. (2001) and in the supplement of Luttkus et al. (2022).'

Line 513:

CSL+OH $\rightarrow$ $\alpha_1$ CV ARO1+ $\alpha_2$ CV ARO2

CSL+NO3 $\rightarrow$ $\alpha_1$ CV ARO1+ $\alpha_2$ CV ARO2

**Reviewer Comment:**

7.) Comparison model-measurement: clearly state that the statistics of the model-observation comparison are given in Table A1. The evaluation should be expanded by calculation of the normalized mean bias (NMB) and FAC2 (fraction of modeled values within factor 2 of measured values). When discussing model underestimation always include the relative bias as NMB (RMSE represents the model error in terms of bias and correlation). The reference to Stern et al. (2008) is not adequate as it refers to PM10 which is much more determined by dust resuspension and Saharan dust events than PM2.5. There are several AQME intercomparison studies which could be cited for discrepancies among models and between modeled and measured concentrations. For PM2.5, different treatment of the formation of secondary aerosols is certainly the most important reason for discrepancies between models. On P10, line 257-259, it is discussed that increased heating and limited mobility caused underestimation of "total pollutants". I would expect that the two activities have opposite effects on certain pollutants, for example NO2 concentration might decrease due to limited mobility whereas PM2.5 concentrations might increase due to more heating in households. The sentence needs to be revised.

**Author's response:**

Thank you for the valuable additions to the model statistical evaluation. We have added the calculated NMB and FAC2 values, and improved the statistical analysis in the results section. Furthermore, we changed the reference to the Im et al. (2015) AQMEII study, which also considers PM2.5 concentrations. We have also amended the relevant sentence.

**Author's changes in manuscript:**

We moved the table with all statistical values from the appendix to the results section (Table 3) and added the new statistics in the text.

Line 153 ff:

'The Normalised Mean Bias (NMB) reflects the systematic bias and indicates a strong underestimation of $PM_{2.5}$ by more than 40% in Melpitz and Frýdlant and - 57% in Košetice.'

Line 258 ff:

'The Root Mean Squared Error (RMSE) quantifies the error between measured and modelled surface-level mass concentrations. Overall, the model RMSE is high with values of 14.26 $\mu gm^{-3}$ for Melpitz, 13.85 $\mu g\ m^{-3}$ for Košetice, and 10.92 $\mu g\ m^{-3}$ for Frýdlant. Together with the NMB, the high RMSE indicate that the model tends to underestimate concentrations during periods of high concentration peaks, as the RMSE is particularly sensitive to outliers. All statistical parameters are presented in Table 3.'

Line 262 ff:

'Im et al. (2015) analysed the performance of multiple models in simulating $PM_{2.5}$ concentrations as part of the AQMEII model intercomparison project. They found that most

models systematically underestimated PM$_{2.5}$ at rural stations, with biases ranging from -2% to -60%. The COSMO-MUSCAT model performed relatively well, showing a bias of -24.82%. However, all models struggled to capture wintertime levels, underestimating concentrations by more than 50% across all regions.'

Line 324 ff:

'Across all three stations, the comparison to the Sunset data show a systematic underestimation by the model, with large negative NMB values: -73% in Melpitz, -79% in Košetice and -67% in Frýdlant.'
* * *
**Reviewer Comment:**

8.) Organic Matter (P 13): Figure 4 shows good agreement among Sunset offline and Sunset online. It should be discussed why OM from Sunset agrees with AMS at Košetice but not at the other sites. Further it should be discussed which of the measurement methods should serve as the guideline for comparison of the modeled OM (OM in PM$_{2.5}$ plus total SOA plus OM from outside the domain). In the text, the terms AMS and ACMS are used interchangeably. It is unclear whether ACMS is an additional instrument or combined with AMS. If it is a separate instrument, why are OM measurements of ACMS not included in Figure 4? At least, it should be made clearer in the text.

**Author's response:**

The discrepancies between AMS/ACSM and Sunset measurements at Melpitz and Frýdlant will be addressed in an upcoming publication by Arora et al. They are multifactorial, primarily driven by variations in aerosol composition and emission sources that influence wintertime measurements. There may be an underestimation in the AMS/ACSM measurements and a simultaneous overestimation in the Sunset data. Since no definitive correction can currently be applied, we report both datasets.

For assessing temporal trends, AMS/ACSM measurements are better suited due to their higher time resolution and finer size cut-off (PM$_1$). This is particularly relevant in winter, when combustion emissions, which primarily fall within the PM$_1$ size range, dominate the aerosol burden. Therefore, it is reasonable to assume that modelled PM$_{2.5}$ concentrations largely consist of PM$_1$ mass, making the comparison reasonable.

On the other hand, the Sunset instrument provides an estimate of the total carbonaceous mass and is useful for assessing the magnitude of concentrations and comparing them directly to PM$_{2.5}$ mass, since it uses the same filters as the gravimetric reference method.

This is why we include and compare both datasets in our analysis, as each provides complementary insights into aerosol composition and concentration.

Regarding instrumentation, the use of different AMS setups at the three sites has been clarified in the text: an ACSM was deployed at Košetice, while standard AMS instruments were used at Melpitz and Frýdlant.

**Author's changes in manuscript:**

Line 110 ff:

'An ACSM(Aerosol Chemical Speciation Monitor) was used for aerosol mass spectrometry at Košetice, while AMS (Aerosol Mass Spectrometer) instruments were used at Melpitz and Frýdlant. Hereafter, we use AMS/ACSM to refer collectively to measurements from all three instruments deployed at the sites.'

Line 328 ff:

'The underestimation of these values by our model seems to have a large contribution to the total $PM_{2.5}$ underestimation. The discrepancy between Sunset and AMS/ACSM observations may partly arise from the different particle size ranges each instrument targets: Sunset samples $PM_{2.5}$, while AMS/ACSM captures only $PM_1$. However, since organic aerosol is predominantly found in the submicrometer size range throughout the year (Poulain et al., 2020), the impact of the size cut-off on the comparison is expected to be minor. This is further supported by observations in Frýdlant, where both $PM_1$ (online) and PM2.5 (offline) Sunset data are available and show only small differences. Nevertheless, other factors contributing to the observed discrepancy cannot be ruled out. AMS/ACSM instruments are particularly well suited for capturing temporal variability, due to their high time resolution. The Sunset instruments provide an estimate of the total carbonaceous mass and are useful for assessing the magnitude of concentrations. It uses the same filters as the gravimetric reference method, allowing a more direct comparison to total $PM_{2.5}$ mass and offering a more complete picture of the aerosol burden.'
* * *
**Reviewer Comment:**

9.) The paragraph on P 14 (line 316-320) should be rephrased. "The discrepancy in modelled PM2.5 concentrations" probably means discrepancy between modeled and measured PM2.5 concentrations. Background PM2.5 is hardly ever driven by elemental carbon since EC concentration are usually rather low (except near sources). Sulfate and nitrate are more likely candidates for mismatch with observed PM2.5, thus the agreement for these components should be stated as well.

**Author's response:**

Thank you very much for your helpful suggestion. We have revised the sentence accordingly and also improved the paragraph discussing sulfate and nitrate. The model tends to overestimate sulfate and nitrate concentrations in Melpitz and Frýdlant, while only Košetice shows a slight underestimation, with average deviations of less than $1\ \mu g\ m^{-3}$. Therefore, it is unlikely that the underestimation of overall $PM_{2.5}$ concentrations by the model is primarily driven by biases in sulfate and nitrate.

**Author's changes in manuscript:**

Line 307 ff:

'At Melpitz, the model performs well for sulfate, with a correlation coefficient of 0.71 and a small bias (NMB = +10%), while nitrate is overestimated (NMB = +51%), though its temporal variability is reasonably captured (R = 0.62) (see Figure 4 panel (a) and (b)). At Frýdlant, the model shows moderate correlations (R = 0.40 - 0.46) and biases (NMB = +29% for sulfate and +40% for nitrate) and a low agreement within a factor of 2 (FAC2 < 50%). Košetice exhibits the weakest agreement, with low correlations (R = 0.16 for nitrate, R = 0.36 for sulfate) and underestimations of both species (NMB = - 26% for nitrate and - 51% for sulfate). These results are broadly in line with model performance criteria reported in the literature, e.g., NMB within 45% for sulfate and   60% for nitrate (Huang et al., 2021), or NMB within ± 30% and R > 0.40 (Emery et al., 2017). This indicates that the model reasonably captures the general magnitude and temporal variability of secondary inorganic aerosol concentrations across the domain, despite some site-specific discrepancies (Table 3). The AMS/ACSM may underestimate total sulfate and nitrate concentrations in winter, when particle growth shifts part of the mass beyond the $PM_1$ range (Poulain et al., 2020), though these species are generally predominantly found in $PM_1$ (Zhang et al., 2023). Given their relatively small contribution to total $PM_{2.5}$ at our sites, it is unlikely that secondary inorganic aerosols are responsible for the discrepancy between the predicted and measured $PM_{2.5}$ aerosol mass concentrations.'
* * *
**Reviewer Comment:**

10.) The numbering and headers of the sections after 3.3 ("Source attribution …") appear to be random and are not well motivated. I suggest bracketing the sections that follow under a "Discussions" chapter (section 4). The second part of section 3.3 could be split off as a discussion section on biomass burning / wood combustion. Together with section 3.4 ("Effects of COVID-19") and section 4 ("anthropogenic secondary organic aerosol") this would form the new discussion chapter.

**Author's response:**

Thank you for this great suggestion! This improves readability significantly, so we have changed the chapter structure accordingly.

**Author's changes in manuscript:**

The chapter structure was changed in the way that was proposed.
* * *
**Reviewer Comment:**

11.) Anthropogenic secondary organic aerosol (P 20): The study of Bergström et al. (2012) was first and should appear first in this section. It would be good to structure the discussion related to publications on SOA modeling in chronological order, as there have

been drastic developments in the treatment of anthropogenic SOA in the last two decades. Can the IVOC emissions be implemented in your model such that IVOC condense to pre-existing OM depending on their volatility or undergo atmospheric aging?

**Author's response:**

We reordered the introduction to the SOA chapter to make it more chronologically consistent.

IVOC are indirectly accounted for in our SOA module SORGAM by production of two oxidation products, a lower and a higher volatility product. However, once formed, the oxidation product does not change chemically, so atmospheric aging is not accounted for.

**Author's changes in manuscript:**

Line 484 ff:

'Bergström et al. (2012) found an underestimation of winter organic aerosol in a modelling study focusing on several years in Europe. Their conclusion was that emissions from wood combustion are under-represented in current emission inventories. Previous source apportionment studies have shown that residential heating is a significant contributor to SOA formation.'
* * *
**Reviewer Comment:**

12.) Sensitivity study results: as in my previous point 8, I wonder which observed metrics / measurement instrument should be used to evaluate changes in mean OM concentrations from the sensitivity tests? On P 22, line 502-510: (a) give absolute OM increment for Melpitz, (b) refer to Table 4 again, (c) discuss that OM from AMS is overestimated with S3 at Frýdlant. In Figure 9, denote error bars for Sunset offline OM which considers instrument uncertainty and OC-factor uncertainty and denote error bars for AMS PM1 data.

**Author's response:**

Thanks for the valuable suggestions. We added measurement errors to the figure and extended the results discussion of the sensitivity studies.

**Author's changes in manuscript:**

Figure 9 was improved by adding error bars.

Line 557 ff:

'This leads to a better agreement with the measurements in Košetice but results in overestimation compared to the AMS/ACSM data at Frýdlant. As discussed previously, the discrepancies between the AMS/ACSM and Sunset measurements cannot be fully resolved in this work, and both datasets must therefore be regarded as valid. Taking into account the

measurement uncertainties, the fact that the simulated OM concentrations at Frýdlant now lie between the two measurements supports the plausibility of the modelled increase. Evaluating both datasets in combination provides a more comprehensive and balanced assessment of actual OM levels. The AMS/ACSM is better suited to capture diurnal patterns due to its higher time resolution. At Frýdlant, the model simulates a clear morning peak in OM concentrations that is absent in the AMS/ACSM data. This discrepancy suggests that the model may be overestimating the contribution from local or near-field sources while underestimating the influence of long-range transport.'
* * *
**Reviewer Comment:**

13.) Sensitivity study results: the discussion of the sensitivity results for CSL emissions and phenol SOA leaves some open questions. Which of the scenarios (base, S1-S3) is now best in reproducing SOA spatial distribution? Figure 10 shows that S3 increases modelled OM in other areas but not around the three study sites. This probably reflects that absorption of SOA to existing PM happens in places where the emissions of PM are already high. This would indicate missing primary OM emissions given the underestimation of measured OM at the sites.

**Author's response:**

Thank you for this valuable remark. We assume that a different spatial distribution of combustion emissions would improve the model performance at our measurement stations, rather than simply increasing primary OM emissions. As the following comment and our corresponding response show, alternative emission inventories that include more detailed information on domestic heating in the Czech Republic exhibit a different spatial emission pattern compared to the inventory used in our study.

**Author's changes in manuscript:**

None
* * *
**Reviewer Comment:**

14.) To test this hypothesis, I suggest to conduct an additional sensitivity test with more detailed residential combustion emission data for the Czech Republic as used in Bartik et al. (2024).

**Author's response:**

An additional sensitivity test using a more detailed emission inventory would be highly valuable. To explore this further, we contacted the authors of the study by Bartik et al.

(2024) and obtained the total annual emissions for the year 2018 for the Czech Republic. This allowed for a spatial comparison with the CAMS emissions used in our model. Unfortunately, a direct implementation of this dataset lies outside the scope of the current revision, as it would require a complex integration with our existing emissions for regions outside the Czech Republic. Moreover, the dataset lacks the specific splitting factors required to convert total $PM_{2.5}$ emissions into OM and EC, which are likely to differ from those used in our current setup.

While the model simulation using the more detailed emission dataset could not be used in the model experiments in this study, the comparison of the emission datasets proved highly informative. We found that the overall annual $PM_{2.5}$ emission flux in the detailed inventory is about 20 % higher than in CAMS. However, more striking is the change in spatial distribution: local differences in total primary $PM_{2.5}$ emissions reach from -100 % to +150 %, with significantly higher emissions in rural areas and lower values in urban regions compared to the CAMS inventory. These findings support our interpretation of underestimated rural emissions and have been incorporated into the conclusion.

[Figure]

**Author's changes in manuscript:**

Line 607 ff:

'A more detailed inventory, as used in Bartík et al. (2024) for the Czech Republic, reveals a redistribution of total primary $PM_{2.5}$ residential combustion emissions from urban to rural areas, compared to the inventory used in this study.'
* * *
Technical corrections:

- P3, line 57: replace "identify PM sources" by "identify primary PM sources".
- P3, line 61: give long name of TRACE and provide web site.
- P5, line 125: resolution should be given in km x km. Same holds for the resolution of CAMS-REG-v5 stated in the next line.
- P12, line 267: replace "below 3 km" by "below 3 km height".
- Figures: captions of Figures 5-10 denote "elemental carbon < PM2.5" or "organic matter < PM2.5". This is not common terminology. Please replace by "in PM2.5" if that is the meaning of "<".

**Author's response:**

These have been corrected. A longer project description including the long name was added in the introduction.

**Author's changes in manuscript:**

Line 63 and 64:

'The tagging approach is applied to identify primary PM sources with a focus on winter combustion emissions.'

Line 58:

'The TRACE project: 'Transport and Transformation of Atmospheric Aerosol over Central Europe with an Emphasis on Anthropogenic Sources' [...]'

There is no project website that we can refer to.

Line 137 ff:

'For European emissions outside Germany the CAMS-REG-v5 emission inventory for the year 2018 (resolution: 6 km x 6 km)'

Line 283 ff:

'Lidar measurements in Leipzig recorded pure dust conditions, but below 3 km height, aerosol from continental Europe was likely mixed into the Saharan dust plumes (Haarig et 285 al., 2022).'

The term < $PM_{2.5}$ was changed to 'in $PM_{2.5}$' in all figures.

---

## Author Comment (AC2)

**Anonymous Referee #2, 03 Jun 2025**

Reviewer comment: *'The manuscript details the use of a non-reactive tagging method of tracking primary organic matter (OM) using the model COSMO-MUSCAT to investigate the contribution of residential heating to OM during the winter of 2021, at 3 measurement stations across central Europe. The findings indicate that the modelled OM is underestimated at these sites, which is mainly attributed to the current under-representation of wood combustion SOA. The study is well framed, and conclusions are adequately presented. I recommend the publication of this work, before the authors clarify and accommodate the following questions/recommendations.'*

**Author's response:**

We thank the reviewer for the careful review of our manuscript and constructive comments and suggestions to improve it. We have modified the manuscript accordingly as outlined below.
* * *
**Reviewer Comment:**

1.) L6:8: "Although the magnitude and temporal changes of the model results mostly agree with total OM values at two measuring stations, it appears to underestimate measurements at a site in the central Czech Republic." It would be better to mention earlier in the introduction that there was 1 measurement site in Germany and 2 in CR to avoid confusion.

**Author's response:**

Good point, we added this information in the abstract.

**Author's changes in manuscript:**

Line 4 and 5:

'The model results are compared with winter measurements from one site in Germany and two sites in the Czech Republic, where solid fuels are commonly used for heating.'
* * *
**Reviewer Comment:**

2.) L18: " wider societal costs associated with it". interesting point but would benefit from stating the wider societal costs. The values can be presented here to underline the magnitude of the costs and losses.

**Author's response:**

We added an estimation of the cost of air pollution:
'A report by the World Bank Group (2022) estimates that the societal cost of ambient fine particulate matter pollution in the Europe and Central Asia region reached 4.6% of gross domestic product (GDP) in 2019. This estimate reflects the economic impact of PM2.5 related health outcomes, including premature mortality, morbidity, and lost productivity'

**Author's changes in manuscript:**

Line 19 ff:

'A report by the World Bank Group (2022) estimates that the societal cost of ambient fine particulate matter pollution in the Europe and Central Asia region reached 4.6% of gross domestic product (GDP) in 2019. This estimate reflects the economic impact of $PM_{2.5}$ related health outcomes, including premature mortality, morbidity, and lost productivity.'
* * *
**Reviewer Comment:**

3.) L71: "The second main wind direction is East (about 17% of the time), with dry continental air masses influenced by long-distance transport from Poland, Belarus, Ukraine, Slovakia and the Czech Republic (Spindler et al., 2001, 2012, 2013).". Is this during summer or winter?

**Author's response:**

The given percentages are referring to the whole year, not only to one season. We changed the sentence accordingly.

**Author's changes in manuscript:**

Line 79 and 80:

'Easterly winds occur 17% of the time throughout the year, bringing dry continental air masses affected by long-range transport from Poland, Belarus, Ukraine, Slovakia, and the Czech Republic (Spindler et al., 2001, 2012, 2013).'
* * *
**Reviewer Comment:**

4.) L135: Add the values of splitting profiles in the appendix.

**Author's response:**

We have added the values to the appendix. For simplicity, we only provide tables for the GNFR C values.

**Author's changes in manuscript:**

Tables A1 and A2 have been added to the appendix. Table A1 provides the splitting factors for particulate matter, while Table A2 provides the splitting factors for non-methane volatile organic compound (NMVOC) emissions.
* * *
**Reviewer Comment:**

5.) L163: wouldn't it benefit to use a 2-d VBS method over the 2 product Odum parameterization?

**Author's response:**

This is a valid point. Implementing the VBS method would enhance the model because it accounts for chemical ageing, unlike our current two-product approach. While incorporating VBS into COSMO-MUSCAT would be complex, it is worth considering for future work. However, this falls outside the scope of this revision.

Nevertheless, we believe that the two-product approach remains suitable for our study. By categorising products into lower- and higher-volatility classes, the approach covers a broad range of SOA products and allows for easy adaptation to new experimental data.

**Author's changes in manuscript:**

None
* * *
**Reviewer Comment:**

6.) L196: add a table in the appendix detailing the main tagged species

**Author's response:**

We have added a table to provide an overview of all the species that have been tagged.

**Author's changes in manuscript:**

Table A3 lists all the species that were tagged in this study.
* * *
**Reviewer Comment:**

7.) L208-209: Since gas phase species and aerosol chemistry is not considered, can the authors quantify how much the SOA will be under-predicted based on the lack of these processes in the model.

**Author's response:**

Our new tagging approach currently only analyses passive tracers (i.e. non-reactive tagging), so SOA is not explicitly evaluated by source region or sector. However, for the given winter scenario we primarily attribute anthropogenic SOA to the other combustion sector (GNFR-C) as the precursor AVOC are mainly emitted from this sector. SOA from anthropogenic sources contributes between 13 - 20 % to the mean total OM mass at the three stations during the investigated wintertime period. Similarly, biogenic SOA contributes 25 - 45 % to total OM on average. Gas-phase processes and aerosol chemistry are generally implemented in COSMO-MUSCAT, the sentence this comment refers to has been revised for better clarity. SOA formation itself is represented via the SORGAM module, which is active in the tagging simulations. However, due to the nonlinearity of SOA formation, it is not possible to directly tag SOA species.

**Author's changes in manuscript:**

Line 224 and 225:

'However, gas phase chemistry and aerosol chemistry are not considered at present within the tagging algorithm.'
* * *
**Reviewer Comment:**

8.) L249: is there a comparison of simulated and measured boundary layer height?

**Author's response:**

Following the other reviewer's suggestion, we have changed this paragraph and removed the citation of Stern et al. (2008) at this point. Consequently, the comment regarding the boundary layer height has been removed.

**Author's changes in manuscript:**

Line 262 ff:

'Im et al. (2015) analysed the performance of multiple models in simulating $PM_{2.5}$ concentrations as part of the AQMEII model intercomparison project. They found that most models systematically underestimated $PM_{2.5}$ at rural stations, with biases ranging from -2%

to -60%. The COSMO-MUSCAT model performed relatively well, showing a bias of -24.82%. However, all models struggled to capture wintertime levels, underestimating concentrations by more than 50% across all regions.'
* * *
**Reviewer Comment:**

9.) L253-254: provide value for the "slight difference".

**Author's response:**

Good point, we changed the sentence to give more information:

**Author's changes in manuscript:**

Line 268 ff:

'The snow event on 7–8 February led to a decrease in $PM_{2.5}$ concentrations in Melpitz by approximately 10 µg m$^{-3}$. In Frýdlant, a slight decrease of around 4 µg m$^{-3}$ was observed after the event, while in Košetice, concentrations even increased by about 4 µg m$^{-3}$, indicating limited overall washout effects.'
* * *
**Reviewer Comment:**

10.) Figure 4: simulated PM2.5 and AMS PM1 are not directly comparable. for e.g if we consider Nitrate how can one interpret high model PM2.5 conc and low AMS PM1 concentrations. Both the values could be in similar if we consider only modelled PM1. Also, 4d is comparing PM1, PM2.5 and PM10. Are the PM1, pm2.5 and PM10 masses correlated? It would perhaps be better to compare, relative or normalized nitrate, sulphate, OM and EC concentrations if one must compare PM1, PM2.5 and PM10 conc.

**Author's response:**

We thank the reviewer for this important feedback. We agree that a direct comparison between modelled $PM_{2.5}$ and observed $PM_1$ (from AMS/ACSM) introduces some uncertainty. However, we believe that the comparison remains valid and informative for multiple reasons.

We acknowledge the findings of Poulain et al. (2020), who noted that ACSM may underestimate total sulfate concentrations when the $PM_1$ fraction of $PM_{2.5}$ mass falls below 60%, as often occurs in winter when particles grow beyond the submicrometer range due to processes like ammonium nitrate condensation. Under such conditions, a portion of sulfate and nitrate mass may reside in particles larger than 1µm and thus be missed by ACSM. This implies that the $PM_{2.5}$ concentrations of sulfate and nitrate may be somewhat

higher than indicated by the ACSM data. Nevertheless, the magnitude of these species remains relatively small at all sites, and their potential uncertainty does not substantially affect our main finding, that the model's underprediction of total $PM_{2.5}$ mass cannot be attributed to a underestimation in inorganic aerosol. Nevertheless, nitrate and sulfate are typically predominantly found in PM1 (Zhang et al., 2023).

We therefore believe that the qualitative comparison between modelled $PM_{2.5}$ and measured $PM_1$ remains informative. The model does not explicitly resolve particle size distributions, and the $PM_{2.5}$ output for these species primarily represents the accumulation mode.
For EC, our model only considers primary combustion sources, which are known to emit particles almost exclusively within the $PM_1$ size range. Observations at Melpitz by Poulain et al. (2011) suggest that more than 90% of eBC mass in $PM_{10}$ is actually in the $PM_1$ fraction. Therefore, comparison of the different size classes should be reasonable. This also applies to the organic aerosol component: organics are mainly distributed in the submicrometer size range throughout the year (Poulain et al. 2020), making the comparison between modelled $PM_{2.5}$ OM and AMS $PM_1$ OM reasonable, especially in winter.

We added this argumentation to our results evaluation in the manuscript.

*Poulain, L., Spindler, G., Birmili, W., Plass-Dülmer, C., Wiedensohler, A., and Herrmann, H.: Seasonal and diurnal variations of particulate nitrate and organic matter at the IfT research station Melpitz, Atmospheric Chemistry and Physics, 11, 12 579–12 599, https://doi.org/doi:10.5194/acp-11-12579-2011, 2011.*

*Poulain, L., Spindler, G., Grüner, A., Tuch, T., Stieger, B., van Pinxteren, D., Petit, J.-E., Favez, O., Herrmann, H., & Wiedensohler, A. (2020). Multi-year ACSM measurements at the central European research station Melpitz (Germany) – Part 1: Instrument robustness, quality assurance, and impact of upper size cutoff diameter. Atmospheric Measurement Techniques, 13, 4973–4994. https://doi.org/10.5194/amt-13-4973-2020*

**Author's changes in manuscript:**

Line 299 ff:

'The AMS/ACSM may underestimate total sulfate and nitrate concentrations in winter, when particle growth shifts part of the mass beyond the $PM_1$ range (Poulain et al., 2020), though these species are generally predominantly found in PM1 (Zhang et al., 2023). Given their relatively small contribution to total $PM_{2.5}$ at our sites, it is unlikely that secondary inorganic aerosols are responsible for the discrepancy between the predicted and measured $PM_{2.5}$ aerosol mass concentrations.'

Line 310 ff:

'Although differences in particle size cut-offs must be considered when comparing observations and model results, Poulain et al. (2011) found that around 90% of the mass of elemental black carbon (eBC) in $PM_{10}$ is contained within the PM1 fraction. Comparing across these different size classes should therefore be reasonable.'

Line 329 ff:

'The discrepancy between Sunset and AMS/ACSM observations may partly arise from the different particle size ranges each instrument targets: Sunset samples $PM_{2.5}$, while AMS/ACSM captures only $PM_1$. However, since organic aerosol is predominantly found in the submicrometer size range throughout the year (Poulain et al., 2020), the impact of the size cut-off on the comparison is expected to be minor. This is further supported by observations in Frýdlant, where both $PM_1$ (online) and $PM_{2.5}$ (offline) Sunset data are available and show only small differences.'
* * *
**Reviewer Comment:**

11.) L297: "The model underestimates the OM concentrations in Košetice (RMSE: 6.48 μg $m^{-3}$) while for Melpitz and Frýdlant the overall fit is good (RMSE: 1.17 and 2.01 μg $m^{-3}$)." doesn't this negate the earlier claim that the OM is underestimated in the simulations?

**Author's response:**

This is correct for AMS/ACSM measurements, but not for those taken using the sunset filter. We have improved the results section and provided a more detailed explanation for each device.

**Author's changes in manuscript:**

Line 324 ff:

'Across all three stations, the comparison to the Sunset data show a systematic underestimation by the model, with large negative NMB values: -73% in Melpitz, -79% in Košetice and -67% in Frýdlant.'

Line 334 ff:

'AMS/ACSM instruments are particularly well suited for capturing temporal variability, due to their high time resolution. The Sunset instruments provide an estimate of the total carbonaceous mass and are useful for assessing the magnitude of concentrations. It uses the same filters as the gravimetric reference method, allowing a more direct comparison to total $PM_{2.5}$ mass and offering a more complete picture of the aerosol burden. In Melpitz and Frýdlant, the model aligns reasonably well with AMS/ACSM observations, with RMSE values of 1.17 and 2.01 μg $m^{-3}$ and NMBs of –8% and +18%, respectively. Correlation is also relatively strong in Melpitz (R = 0.60), but lower in Frýdlant (R = 0.19), where the model fails to capture diurnal variability. The model underestimates the OM concentrations by AMS/ACSM in Košetice (RMSE: 6.48 μg $m^{-3}$; NMB: –74%) and also does not fully reproduce the diurnal variations (R = 0.39) (see Fig. A2 in the Appendix).'

**Reviewer Comment:**

12.) L310-315: Since the winter values are heavily meteorological dependent one must mention if the weather patterns during the said years matched 2021. From the description above the year 2021 seems to be an odd one considering the long Sahara dust events and the cold periods. I would suggest making such comparisons to more tangible SOA concentrations, which can then present a broader picture of a trend in SOA underestimation.

**Author's response:**

It is true, that the direct comparison with previous years is not directly possible, but still it might give some valuable information about the study sites. We therefore would like to keep the values in the manuscript, we added an explanatory sentence to make the differences clear.

**Author's changes in manuscript:**

Line 363 ff:

'For our study period we found Sunset Filter values ranging in average from 5.06 µg m$^{-3}$ in Melpitz to 7.74 µg m$^{-3}$ in Košetice, exceeding typical values reported for previous years. This suggests a strong influence of meteorological conditions on the overall concentration levels.'
* * *
**Reviewer Comment:**

13.) L 345: Public power contribution. This is interesting. One would expect higher contributions from public power at Košetice especially in the cold period as the air masses is stagnant. Can you explain why is the contribution of public power low during the cold period?

**Author's response:**

Yes, we would assume that the relevant power plants are not close enough. Overall, Public Power's contributions are low at all stations. For Frýdlant, the proximity to the Turów power plant can be seen in the changing country contributions.

**Author's changes in manuscript:**

None
* * *
**Reviewer Comment:**

14.) L 358-359: Melpitz cross border transport. Is this during the cold or warm period? Looking at Figure 7, it appears that Košetice and Frýdlant have larger cross-border (Poland) contribution to public power than Melpitz. Also, it appears that at Melpitz the cross border contibutions is more in the warm period but at kosetic the polish and german contributions are significant even during the cold period.

**Author's response:**

The high cross-border contributions in Melpitz refer only to the 'Other Combustion' sector. The public power sector also exhibits higher levels of cross-border pollution in Košetice and Frýdlant. The relevant sentence has been revised for clarity.

**Author's changes in manuscript:**

Line 405 ff:

'Contributions to fine OM from the 'other Combustion' sector are highest in the Czech Republic and in urban agglomerations in Poland and around Berlin, Germany (see Fig. 8, right panel). The main contributors to the concentrations observed at the stations are emissions originating within the country where the station is located.'
* * *
**Reviewer Comment:**

15.) Figure 7: I would suggest removing the wind barbs since it doesn't add any information's. or did the authors miss the y axis with the degrees?

**Author's response:**

We consider the correlation between fluctuations in wind direction and alterations in country contributions to be a subject of interest. Therefore, the windbarbs provide information on the prevailing wind at that time. To provide further clarification, the windbarbs have been included in the legend. The orientation of the barbs indicates the wind direction. We therefore assume that an additional y-axis is not necessary.

**Author's changes in manuscript:**

We have included a description of the wind barbs in the legend for Figure 7.
* * *
**Reviewer Comment:**

16.) Table 3: why is alpha 1 same for S1 and S3? Shouldn't it be same for S2 and S3? Some explanation is needed in the main body or the table caption.

**Author's response:**

S1 only includes the SOA yield changes, which is done by adjusting alpha 1. While S2 only includes the additional emissions with SOA yields as in the base run. S3 then includes both. We added more detailed explanation in the table description

**Author's changes in manuscript:**

The table referred to here is now Table 4. We have improved the description:
'Table 4. Overview of the sensitivity simulations. Shown are changes to $\alpha_1$ to adjust the SOA yield parameter for aromatic precursors and scaling of CSL emissions based on CO emissions from GNFR C to account for phenol contributions.'
* * *
**Reviewer Comment:**

17.) L496-497: it would be interesting to see if the increase in OM holds true during the cold period of stagnant airmass at these sites.

**Author's response:**

The underestimation during this period is substantial and cannot be attributed solely to the underprediction of SOA. The increased emissions used in our sensitivity simulation do not fully explain the significant discrepancy. Therefore, it would be highly beneficial to incorporate temperature-dependent combustion emissions in future simulations to better capture increased heating activity during cold periods. This is planned as the next step to further improve model performance.

**Author's changes in manuscript:**

None
* * *
**Reviewer Comment:**

18.) L 537-538: I would suggest saying that the diurnal profiles are reproduced, not the total OM magnitude.

**Author's response:**

Yes! This is true since the AMS data is represented well but Sunset data is underestimated at all stations. We changed the sentence accordingly.

**Author's changes in manuscript:**

Line 597 ff:

'Although the present study reproduced diurnal OM profiles well at two monitoring sites, measurements at Košetice are underestimated, partly due to an inadequate representation of SOA formation from residential heating (wood combustion), a major source of anthropogenic VOCs.'